# Autophagy compensates for defects in mitochondrial dynamics

**Simon Haeussler**[1Ⓨ], **Fabian Köhler**[1Ⓨ], **Michael Witting**[2,3], **Madeleine F. Premm**[1], **Stéphane G. Rolland**[1], **Christian Fischer**[1,4], **Laetitia Chauve**[5], **Olivia Casanueva**[5], **Barbara Conradt**[1,4,6]*

**1** Faculty of Biology, Ludwig-Maximilians-University Munich, Munich, Germany, **2** Research Unit Analytical BioGeoChemistry, Helmholtz Zentrum München, Neuherberg, Germany, **3** Chair of Analytical Food Chemistry, Technische Universität München, Freising, Germany, **4** Center for Integrated Protein Science, Ludwig-Maximilians-University Munich, Planegg-Martinsried, Germany, **5** Epigenetics Programme, The Babraham Institute, Cambridge, United Kingdom, **6** Department of Cell and Developmental Biology, Division of Biosciences, University College London, London, United Kingdom

Ⓨ These authors contributed equally to this work.

* b.conradt@ucl.ac.uk

**Data Availability Statement:** All relevant data are within the manuscript and its Supporting Information files.

**Funding:** This work was supported by the Deutsche Forschungsgemeinschaft https://eur01.

## Abstract

Compromising mitochondrial fusion or fission disrupts cellular homeostasis; however, the underlying mechanism(s) are not fully understood. The loss of *C. elegans fzo-1*[MFN] results in mitochondrial fragmentation, decreased mitochondrial membrane potential and the induction of the mitochondrial unfolded protein response (UPR[mt]). We performed a genome-wide RNAi screen for genes that when knocked-down suppress *fzo-1*[MFN](lf)-induced UPR[mt]. Of the 299 genes identified, 143 encode negative regulators of autophagy, many of which have previously not been implicated in this cellular quality control mechanism. We present evidence that increased autophagic flux suppresses *fzo-1*[MFN](lf)-induced UPR[mt] by increasing mitochondrial membrane potential rather than restoring mitochondrial morphology. Furthermore, we demonstrate that increased autophagic flux also suppresses UPR[mt] induction in response to a block in mitochondrial fission, but not in response to the loss of *spg-7*[AFG3L2], which encodes a mitochondrial metalloprotease. Finally, we found that blocking mitochondrial fusion or fission leads to increased levels of certain types of triacylglycerols and that this is at least partially reverted by the induction of autophagy. We propose that the breakdown of these triacylglycerols through autophagy leads to elevated metabolic activity, thereby increasing mitochondrial membrane potential and restoring mitochondrial and cellular homeostasis.

## Author summary

Various quality control mechanisms within the cell ensure mitochondrial homeostasis. Specifically, mitochondrial fission and fusion, the mitochondrial unfolded protein response (UPR[mt]) and/or mitophagy are induced upon mitochondrial stress to maintain or restore mitochondrial homeostasis. How these different quality control mechanisms are coordinated and how they influence each other is currently not well understood.

safelinks.protection.outlook.com/?url=https%3A%2F%2Fwww.dfg.de%2Fen%2F&data=02%7C01%7C%7C381f38b59bcd4e9a567f08d7bba8aec2%7C1faf88fea9984c5b93c9210a11d9a5c2%7C0%7C0%7C637184205470464132&sdata=Eb4roXJqDx7UamjpWqYooqGw6RKzo5yD8jDFkLtapls%3D&reserved=0 (CO204/6-1, CO204/9-1 and EXC114 to BC). Some strains were provided by the CGC, which is funded by NIH Office of Research Infrastructure Programs https://eur01.safelinks.protection.outlook.com/?url=https%3A%2F%2Fwww.nih.gov%2F&data=02%7C01%7C%7C381f38b59bcd4e9a567f08d7bba8aec2%7C1faf88fea9984c5b93c9210a11d9a5c2%7C0%7C0%7C637184205470464132&sdata=LGLqn8cL5mhk9Xn6gWttZPV5hOmsiv%2Fi13%2BC6CDBYPU%3D&reserved=0 (P40 OD010440). The funders had no role in study design, data collection and analysis, decision to publish, or preparation of the manuscript.

**Competing interests:** The authors have declared that no competing interests exist.

Interestingly, the disruption of mitochondrial dynamics has recently been shown to induce UPR$^{mt}$. We performed a genome-wide RNAi screen for suppressors of UPR$^{mt}$ induced by a block in mitochondrial fusion and found approximately half of the candidate genes identified to negatively regulate autophagy, a central quality control mechanism that adjusts cellular metabolism under conditions of stress. Furthermore, we found that induction of autophagy also suppresses UPR$^{mt}$ induced by a block in mitochondrial fission. In addition, we demonstrate that defects in mitochondrial dynamics lead to changes in lipid metabolism, which can partially be reverted by the induction of autophagy. Taken together, our results suggest a so far unknown functional connection between UPR$^{mt}$ and autophagy in animals with defects in mitochondrial dynamics.

## Introduction

Mitochondrial dynamics plays an important role in the maintenance of mitochondrial function and, hence, cellular homeostasis [1]. Mitochondrial fission and fusion are both mediated by members of the family of dynamin-like guanosine triphosphatases (GTPases) [2]. In the nematode *Caenorhabditis elegans*, mitochondrial fission is facilitated by the cytosolic dynamin-like GTPase DRP-1$^{DRP1}$, which is recruited to mitochondria where it presumably forms constricting spirals as shown for its *Saccharomyces cerevisiae* counterpart Drp1 [3,4]. Conversely, fusion of the outer and inner mitochondrial membranes is carried out by the membrane-anchored dynamin-like GTPases FZO-1$^{MFN}$ [5] and EAT-3$^{OPA1}$ [6], respectively. The consequences with respect to mitochondrial function and cellular homeostasis of disrupting mitochondrial dynamics are not yet fully understood; however, it has recently been demonstrated that this activates a retrograde quality control signaling pathway referred to as the 'mitochondrial Unfolded Protein Response' (UPR$^{mt}$) [7,8]. In *C. elegans*, UPR$^{mt}$ is activated upon mitochondrial stress, which leads to a decrease in mitochondrial membrane potential and the subsequent import into the nucleus of the 'Activating Transcription Factor associated with Stress 1' (ATFS-1$^{ATF4,5}$) [9,10]. ATFS-1$^{ATF4,5}$ harbors both an N-terminal mitochondrial targeting sequence and a C-terminal nuclear localization sequence and is normally imported into mitochondria [11]. Upon mitochondrial stress, ATFS-1$^{ATF4,5}$ is imported into the nucleus, where it cooperates with the proteins UBL-5$^{UBL5}$ and DVE-1$^{SATB1}$ to promote the transcription of genes that act to restore mitochondrial function and to adjust cellular metabolism [9,10,12,13]. Among these genes are the mitochondrial chaperone genes *hsp-6*$^{mtHSP70}$ and *hsp-60*$^{HSP60}$, the transcriptional upregulation of which is commonly used to monitor UPR$^{mt}$ activation [14].

Whereas UPR$^{mt}$ is a quality control pathway that is activated upon mitochondrial stress, macro-autophagy (from now on referred to as 'autophagy') is a more general cellular quality control mechanism. Through autophagy, cytosolic constituents, long-lived proteins or dysfunctional organelles are degraded and recycled [15,16]. Upon the induction of autophagy, a double-membrane structure called 'phagophore' forms, which enlarges and eventually engulfs the cargo to form an 'autophagosome'. The autophagosome then fuses with a lysosome to form an 'autolysosome', in which the engulfed cargo is subsequently degraded by lysosomal hydrolases [16–18]. A key regulator of autophagy in *C. elegans* is the kinase LET-363$^{mTOR}$ [19]. When cellular nutrients are abundant, LET-363$^{mTOR}$ represses the 'induction complex', which includes UNC-51$^{ULK}$, a kinase that initiates autophagy [20–26].

Another vesicular process that targets cargo for degradation to the lysosome is endocytosis. The 'Endosomal Sorting Complex Required for Transport' (ESCRT) plays a critical role in

endocytosis [27,28]. The ESCRT is composed of five different subcomplexes (ESCRT-0, -I, -II, -III and the AAA-ATPase VPS4) and was originally identified because of its role in the formation of multivesicular bodies (MVBs), which enables ubiquitinated membrane proteins to be sorted into small intralumenal vesicles (ILVs) [29,30]. The ESCRT has since been shown to be required for a number of other cellular processes, such as cytokinesis and virus budding [27,31,32]. ESCRT activity has also been shown to affect autophagy. Studies in mammals and *Drosophila melanogaster* demonstrated that depleting ESCRT components results in a block in autophagy and that in these animals, the ESCRT is required for the fusion of endosomes with lysosomal compartments and also autophagosomes [33–36]. Moreover, ESCRT components have recently been shown to be involved in the closure of autophagosomes in mammals and yeast [37,38]. However, in *C. elegans*, the depletion of ESCRT components results in the induction of autophagy, which suggests that in this species, ESCRT function antagonizes or suppresses autophagy [39,40].

Whereas a functional connection between the ESCRT and autophagy has been established in yeast, nematodes, flies and mammals [33–40], functional connections between the ESCRT and UPR$^{mt}$ or between autophagy and UPR$^{mt}$ [40] have not been described or are poorly understood. In this study, we present evidence that in *C. elegans*, the ESCRT, autophagy and UPR$^{mt}$ functionally interact. Specifically, we found that the induction of autophagy suppresses UPR$^{mt}$ induced by a block in mitochondrial fusion or fission. Interestingly, lipid profiling revealed alterations in the lipidome of mutants defective in mitochondrial dynamics, and we present evidence that changes in the levels of certain types of triacylglycerols (TGs) in *fzo-1*$^{MFN}$ mutants can be reverted by the induction of autophagy. We propose that through the breakdown of these triacylglycerols, the induction of autophagy leads to elevated metabolic activity, thereby increasing mitochondrial membrane potential and restoring mitochondrial and, hence, cellular homeostasis.

## Results

In *C. elegans*, knock-down by RNA-mediated interference (RNAi) of genes encoding dynamin-like GTPases required for mitochondrial fusion (*fzo-1*$^{MFN}$, *eat-3*$^{OPA1}$) or mitochondrial fission (*drp-1*$^{DRP1}$) induces the 'mitochondrial Unfolded Protein Response' (UPR$^{mt}$) [7,8]. Using a multi-copy transgene of the transcriptional reporter P$_{hsp-6\ mtHSP70}$*gfp* (*zcIs13*) [14], we tested strong loss-of-function (lf) mutations of *fzo-1*$^{MFN}$ and *drp-1*$^{DRP1}$ (*fzo-1(tm1133)*, *drp-1 (tm1108)* (National BioResource Project)) and found that they induce UPR$^{mt}$ to different degrees (S1A and S1C Fig). As a positive control, we used animals carrying a lf mutation of the gene *spg-7*$^{AFG3L2}$ (*spg-7(ad2249)*), which encodes a mitochondrial metalloprotease required for mitochondrial function [41]. The *zcIs13* transgene shows very low baseline expression in wild-type animals and is widely used to monitor UPR$^{mt}$ in *C. elegans* [7,9–14,42–44]. In the case of *fzo-1(tm1133)* animals, for example, its expression is induced more than 15-fold (S1C Fig). Furthermore, RNAi knock-down of *spg-7*$^{AFG3L2}$ or genes encoding subunits of the electron transport chain (ETC), or treatments with drugs targeting the latter (e.g. antimycin) lead to strong induction of *zcIs13* expression [14,43]. This makes the *zcIs13* transgene suitable for high throughput, large-scale screens.

However, considering that *fzo-1(tm1133)* causes an increase in the amount of endogenous HSP-6$^{mtHSP70}$ protein by only 1.44-fold (S1E Fig), the fold induction observed with the multi-copy *zcIs13* transgene may not reflect the physiological response with respect to UPR$^{mt}$ induction by the loss of *fzo-1*$^{MFN}$. Furthermore, the *zcIs13* transgene exhibits large variability in expression between animals (inter-individual variability) (S1A Fig), which makes it difficult to obtain consistent results, especially when knocking-down genes using RNA-mediated

interference (RNAi). For this reason, we generated a single-copy transgene, *bcSi9* (integrated at a defined chromosomal location using MosSCI), of the transcriptional reporter P*hsp-6* mtHSP70*gfp*. As shown in S1B Fig, the *bcSi9* transgene shows low baseline expression and, in the case of *spg-7 (ad2249)* and *fzo-1(tm1133)*, an increase in expression of ~5-fold or ~4-fold, respectively (S1D Fig). Furthermore, compared to *fzo-1(tm1133)* animals carrying the multi-copy transgene *zcIs13*, *fzo-1(tm1133)* animals carrying the single-copy transgene *bcSi9* exhibit less inter-individual variability (S1A and S1B Fig). Similarly, *drp-1(tm1108)* animals carrying *bcSi9* show significantly less inter-individual variability compared to *drp-1(tm1108)* animals carrying the multi-copy transgene *zcIs13* (S1A and S1B Fig). Importantly, for all genotypes tested, we found that compared to the fold-induction observed with the multi-copy transgene *zcIs13*, the fold-induction observed with the single-copy transgene *bcSi9* correlated better with the fold-induction observed in the amount of endogenous HSP-6$^{mtHSP70}$ protein (S1A–S1E Fig). Finally, to compare inter-individual variability of the expression of the two P*hsp-6* mtHSP70*gfp* transgenes *zcIs13* and *bcSi9* as well as the endogenous *hsp-6*$^{mtHSP70}$ locus in a quantitative manner, we performed single-worm RT-qPCR experiments in synchronized populations of 36 individual animals and compared inter-individual variability in expression of *zcIs13*, *bcSi9* or the endogenous *hsp-6*$^{mtHSP70}$ locus to those of loci with low (*hsp-1*$^{HSPA1L}$), medium (*ttr-45*) or high (*nlp-29*) inter-individual variability in expression, respectively (S1F Fig). While the expression of the endogenous *hsp-6*$^{mtHSP70}$ locus is not variable between individuals of a population, the expression of the multi-copy transgene *zcIs13* is highly variable in both a wild-type and *fzo-1(tm1133)* background (S1F Fig). Furthermore, the single-copy transgene *bcSi9* exhibits some inter-individual variability in expression, however, to a much lower degree than the transgene *zcIs13*. Therefore, based on these results, we decided to use the multi-copy transgene *zcIs13* for a genome-wide RNAi screen for suppressors of *fzo-1(tm1133)*-induced UPR$^{mt}$ and the single-copy transgene *bcSi9* for subsequent analyses of candidates identified (see below).

## Depletion of ESCRT components suppresses *fzo-1(tm1133)*-induced UPR$^{mt}$

To identify genes that affect the induction of UPR$^{mt}$ in response to a block in mitochondrial fusion, we performed a genome-wide RNAi screen using *fzo-1(tm1133)* animals carrying the multi-copy P*hsp-6* mtHSP70*gfp* transgene *zcIs13* (S1A Fig). To that end, we used an RNAi feeding library that covers approximately 87% of *C. elegans* protein coding genes [45] and analyzed animals of the F1 generation. Among the 299 suppressors identified, three genes, *vps-4*$^{VPS4}$, *vps-20*$^{CHMP6}$ and *vps-37*$^{VPS37}$, encode components of the 'Endosomal Sorting Complex Required for Transport' (ESCRT) [27–30]. We analyzed the suppression of *fzo-1(tm1133)*-induced UPR$^{mt}$ using the single-copy P*hsp-6* mtHSP70*gfp* transgene *bcSi9* and found that knock-down of *vps-4*$^{VPS4}$ or *vps-20*$^{CHMP6}$ by RNAi (referred to as '*vps-4(RNAi)*' or '*vps-20(RNAi)*') causes suppression by 39% or 23% on average, respectively (Fig 1A and 1C). *vps-37(RNAi)* does not result in a statistically significant suppression on average; however, some individual animals show strong suppression (see Fig 1A; *vps-37(RNAi)*; red arrowheads). As a positive control, we knocked-down the function of *atfs-1*$^{ATF4,5}$ by RNAi, which results in suppression of *fzo-1(tm1133)*-induced UPR$^{mt}$ by 54% on average. (In a wild-type background, *atfs-1 (RNAi)*, *vps-4(RNAi)* or *vps-20(RNAi)* suppresses baseline expression of the *bcSi9* transgene by 8%, 14% or 14%, respectively (S2A Fig).) To confirm the suppression of *fzo-1(tm1133)*-induced UPR$^{mt}$ upon *ESCRT(RNAi)*, we used a multi-copy transgene *(zcIs9)* of a transcriptional reporter of the gene *hsp-60*$^{HSP60}$ (P*hsp-60* HSP60*gfp)*, which is also transcriptionally upregulated in response to the induction of UPR$^{mt}$ [14]. Using the P*hsp-60* HSP60*gfp* reporter, we found that *vps-37(RNAi)*, *vps-20(RNAi)* or *vps-4(RNAi)* suppresses by 34%, 41% or 33% on average, respectively (Fig 1B and 1D).

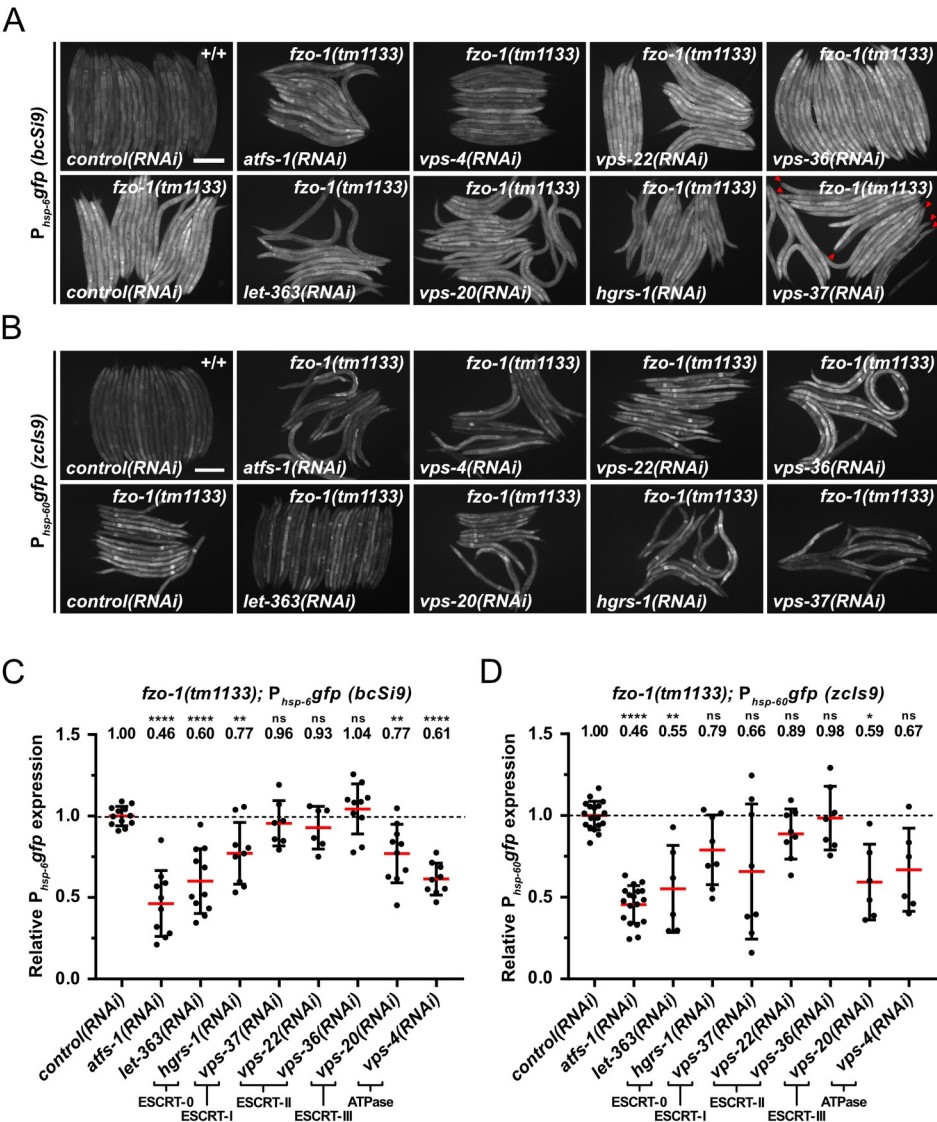

**Fig 1. Depletion of ESCRT components and LET-363 suppresses *fzo-1(tm1133)*-induced UPR^mt. (A)** Fluorescence images of L4 larvae expressing P_{hsp-6}*gfp (bcSi9)* in wild type (+/+) or *fzo-1(tm1133)*. L4 larvae were subjected to *control (RNAi)*, *atfs-1(RNAi)*, *vps-4(RNAi)*, *vps-20(RNAi)*, *vps-22(RNAi)*, *hgrs-1(RNAi)*, *vps-36(RNAi)*, *vps-37(RNAi)* or *let-363 (RNAi)* and the F1 generation was imaged. Red arrowheads indicate suppressed animals upon *vps-37(RNAi)*. Scale bar: 200 μm. **(B)** Fluorescence images of L4 larvae expressing P_{hsp-60}*gfp (zcIs9)* in wild type (+/+) or *fzo-1(tm1133)*. L4 larvae were subjected to *control(RNAi)*, *atfs-1(RNAi)*, *vps-4(RNAi)*, *vps-20(RNAi)*, *vps-22(RNAi)*, *hgrs-1(RNAi)*, *vps-36 (RNAi)*, *vps-37(RNAi)* or *let-363(RNAi)* and the F1 generation was imaged. Scale bar: 200 μm. **(C)** Quantifications of fluorescence images from panel A. After subtracting the mean fluorescence intensity of wild type (+/+) on *control (RNAi)*, the values were normalized to *fzo-1(tm1133)* on *control(RNAi)*. Each dot represents the quantification of fluorescence intensity of 15–20 L4 larvae. Values indicate means ± SD of at least 3 independent experiments in duplicates. $^{**}P<0.01$, $^{****}P<0.0001$ using one-way ANOVA with Dunnett's multiple comparison test to *control (RNAi)*. **(D)** Quantifications of fluorescence images from panel B. After subtracting the mean fluorescence intensity of wild type (+/+) on *control(RNAi)*, the values were normalized to *fzo-1(tm1133)* on *control(RNAi)*. Each dot represents the quantification of fluorescence intensity of 10–20 L4 larvae. Values indicate means ± SD of 3 independent experiments in duplicates. ns: not significant, $^{*}P<0.05$, $^{**}P<0.01$, $^{****}P<0.0001$ using Kruskal-Wallis test with Dunn's multiple comparison test to *control(RNAi)*.

To validate that the reduced P_{hsp-6 mtHSP70}*gfp (bcSi9)* and P_{hsp-60 HSP60}*gfp (zcIs9)* expression in *fzo-1(tm1133)* animals upon *ESCRT(RNAi)* is specific to the UPR^mt response, we tested a

transcriptional reporter, P$_{ges-1\ CES2}$gfp, that has a similar expression pattern as the two UPR$^{mt}$ reporters. Depletion of ESCRT component VPS-4$^{VPS4}$ or VPS-20$^{CHMP6}$ does not result in suppression of the P$_{ges-1\ CES2}$gfp reporter (Fig 2A and 2B), suggesting that ESCRT depletion does not cause degradation of cytosolic GFP *per se* but specifically suppresses the expression of the two UPR$^{mt}$ reporters.

Since *vps-4*$^{VPS4}$, *vps-20*$^{CHMP6}$ and *vps-37*$^{VPS37}$ are part of different ESCRT subcomplexes (*vps-4*$^{VPS4}$—ATPase, *vps-20*$^{CHMP6}$—ESCRT-III, *vps-37*$^{VPS37}$—ESCRT-I) [27], we tested whether depletion of components of the two remaining ESCRT subcomplexes, ESCRT-0 and ESCRT-II, also suppresses *fzo-1(tm1133)*-induced UPR$^{mt}$. Using the P$_{hsp-6\ mtHSP70}$gfp reporter *(bcSi9)*, we found that RNAi knock-down of *hgrs-1*$^{HGS}$ (ESCRT-0) suppresses by 23% on average (Fig 1A and 1C). In contrast, RNAi knock-down of two genes encoding components of ESCRT-II, *vps-22*$^{SNF8}$ and *vps-36*$^{VPS36}$, fails to suppress. Similarly, using the P$_{hsp-60\ HSP60}$gfp reporter *(zcIs9)*, we found suppression by *hgrs-1(RNAi)* but not *vps-22(RNAi)* or *vps-36(RNAi)* (Fig 1B and 1D). Taken together, our results demonstrate that the depletion of components of ESCRT-0, -I, -III or VPS-4 ATPase can suppress *fzo-1(tm1133)-*induced UPR$^{mt}$.

## Depletion of ESCRT components does not rescue the fragmented mitochondria phenotype in *fzo-1(tm1133)* animals but increases mitochondrial membrane potential

The loss of *fzo-1*$^{MFN}$ function has a dramatic effect on steady-state mitochondrial morphology. This is easily detectable in *C. elegans* body wall muscles using a reporter that drives the expression of mitochondrial-matrix targeted GFP protein *(*P$_{myo-3\ MYH}$gfp$^{mt}$*)* [3,5,46]. In *control (RNAi)* animals, the mitochondria in body wall muscle cells are predominantly tubular (Fig 2C). In contrast, in *fzo-1(tm1133)* animals treated with *control(RNAi)*, the mitochondria are predominantly fragmented (referred to as 'fragmented mitochondria' phenotype). To determine whether the depletion of components of ESCRT-I or -III, or the depletion of the ATPase VPS-4$^{VPS4}$ restores steady-state mitochondrial morphology, we analyzed mitochondrial morphology in *fzo-1(tm1133)* animals, in which *vps-4*$^{VPS4}$, *vps-20*$^{CHMP6}$ or *vps-37*$^{VPS37}$ had been knocked-down by RNAi. We found that knock-down of these genes has no effect on the fragmented mitochondria phenotype in body wall muscle cells of *fzo-1(tm1133)* animals (Fig 2C). Knock-down of *vps-4*$^{VPS4}$, *vps-20*$^{CHMP6}$ or *vps-37*$^{VPS37}$ in *fzo-1(tm1133)* animals also has no effect on mitochondrial morphology in hypodermal or intestinal cells (Fig 2E and S3B Fig). (ESCRT depletion has no effect on steady-state mitochondrial morphology in body wall muscle cells in a wild-type background (S3A Fig).)

Since we did not see a change in mitochondrial morphology in *fzo-1(tm1133)* animals upon *ESCRT(RNAi)*, we tested whether it affects mitochondrial membrane potential. Therefore, we stained larvae with TMRE (Tetramethylrhodamine ethyl ester), a membrane potential dependent dye that is commonly used in *C. elegans* to measure mitochondrial membrane potential in hypodermal cells [10,14]. To measure the intensity of TMRE signal, mitochondria in the fluorescent images were segmented using Fiji image software to generate a binary mask (S4 Fig). This mask, which includes all mitochondria of an image, was then used to measure TMRE fluorescence intensity per mitochondrial area in the raw image. Compared to wild type, TMRE fluorescence intensity per mitochondrial area was reduced by 63% in *fzo-1 (tm1133)* animals (Fig 2D). We found increased levels of TMRE fluorescence intensity per mitochondrial area in *fzo-1(tm1133)* animals upon *vps-4(RNAi)* (19%) or *vps-20(RNAi)* (33%), compared to *control(RNAi)* (Fig 2E and 2F). In contrast, ESCRT depletion in the wild-type background causes a reduction in TMRE fluorescence intensity per mitochondrial area by 24% upon *vps-4(RNAi)* or 18% upon *vps-20(RNAi)* (Fig 2G and 2H). Mitochondrial TMRE

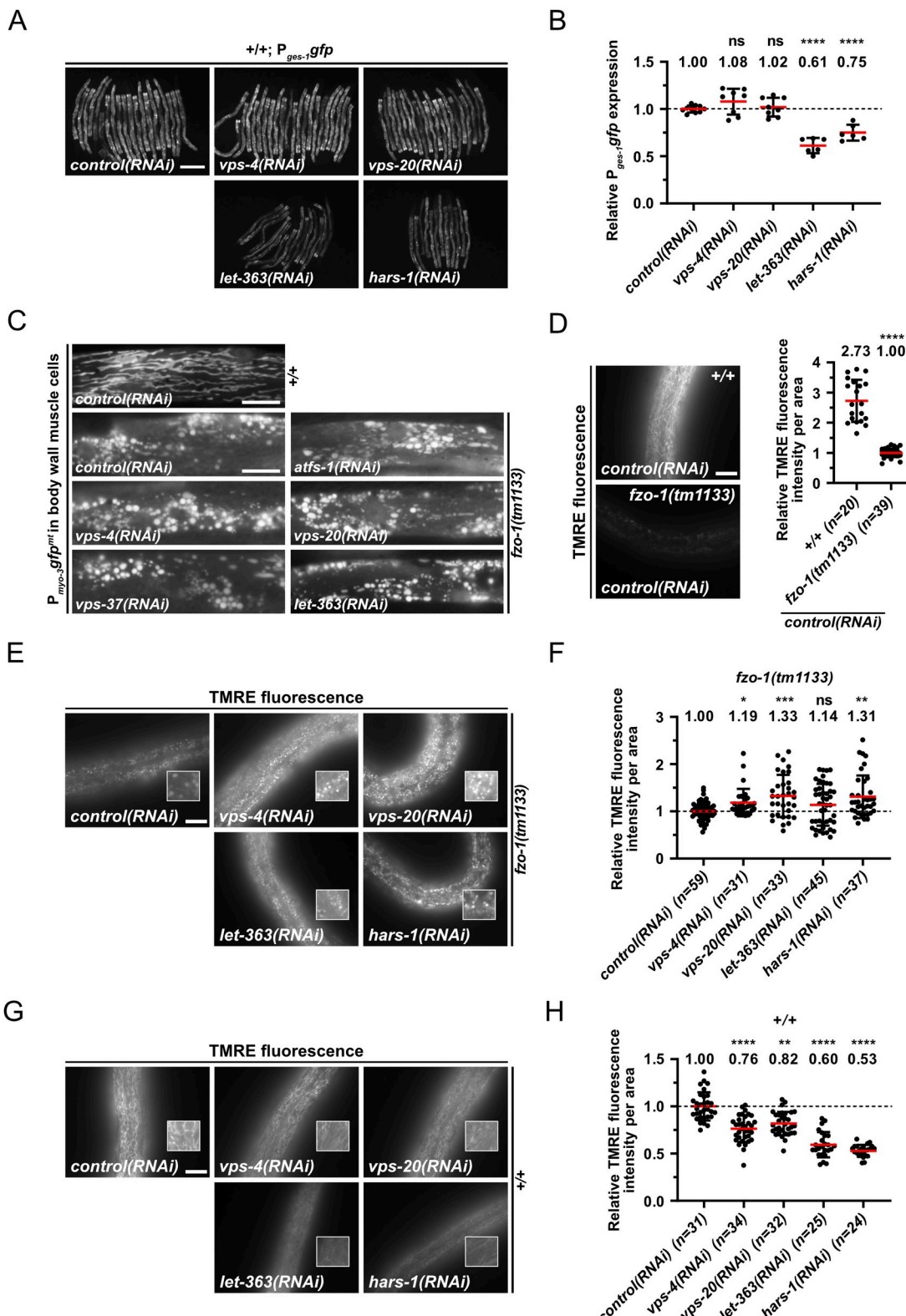

**Fig 2. Induction of autophagy increases mitochondrial membrane potential and suppresses *fzo-1(tm1133)*-induced UPR^mt.** (**A**) Fluorescence images of L4 larvae expressing P_*ges-1*gfp in wild type (+/+). L4 larvae were subjected to *control(RNAi)*, *vps-4 (RNAi)*, *vps-20(RNAi)*, *let-363(RNAi)* or *hars-1(RNAi)* and the F1 generation was imaged. Scale bar: 200 μm. (**B**) Quantifications of fluorescence images from panel A. The values were normalized to *control(RNAi)* and each dot represents the quantification of

fluorescence intensity of 15–20 L4 larvae. Values indicate means ± SD of 3 independent experiments in duplicates. ns: not significant, ****$P<0.0001$ using one-way ANOVA with Dunnett's multiple comparison test to *control(RNAi)*. (C) Fluorescence images of L4 larvae expressing P_myo-3_*gfp*^mt in wild type (+/+) or *fzo-1(tm1133)*. L4 larvae were subjected to *control(RNAi)*, *atfs-1 (RNAi)*, *vps-4(RNAi)*, *vps-20(RNAi)*, *vps-37(RNAi)* or *let-363(RNAi)* and the F1 generation was imaged. Scale bar: 10 μm. (D) Fluorescence images and quantifications of L4 larvae stained with TMRE in wild type (+/+) or *fzo-1(tm1133)*. L4 larvae were subjected to *control(RNAi)* and the F1 generation was stained with TMRE overnight and imaged. Scale bar: 10 μm. Values indicate means ± SD of 3 independent experiments in duplicates. ****$P<0.0001$ using unpaired two-tailed t-test with Welch's correction. (E) Fluorescence images of L4 larvae stained with TMRE in *fzo-1(tm1133)*. L4 larvae were subjected to *control(RNAi)*, *vps-4(RNAi)*, *vps-20(RNAi)*, *let-363(RNAi)* or *hars-1(RNAi)* and the F1 generation was stained with TMRE overnight and imaged. Scale bar: 10 μm. (F) Quantifications of fluorescence images from panel E. The values were normalized to *fzo-1(tm1133)* on *control(RNAi)* and each dot represents the quantification of fluorescence intensity per area from one L4 larvae. Values indicate means ± SD of 3 independent experiments in duplicates. ns: not significant, *$P<0.05$, **$P<0.01$, ***$P<0.001$ using Kruskal-Wallis test with Dunn's multiple comparison test to *control(RNAi)*. (G) Fluorescence images of wild-type L4 larvae stained with TMRE. L4 larvae were subjected to *control(RNAi)*, *vps-4(RNAi)*, *vps-20(RNAi)*, *let-363(RNAi)* or *hars-1(RNAi)* and the F1 generation was stained with TMRE overnight and imaged. Scale bar: 10 μm. (H) Quantifications of fluorescence images from panel G. The values were normalized to wild type on *control(RNAi)* and each dot represents the quantification of fluorescence intensity per area from one L4 larvae. Values indicate means ± SD of 3 independent experiments in duplicates. **$P<0.01$, ****$P<0.0001$ using Kruskal-Wallis test with Dunn's multiple comparison test to *control(RNAi)*.

fluorescence intensity is proportional to mitochondrial membrane potential [47]. Therefore, *ESCRT(RNAi)* results in an increase in mitochondrial membrane potential in *fzo-1(tm1133)* mutants. Hence, our data suggests that the suppression of *fzo-1(tm1133)*-induced UPR^mt upon ESCRT depletion is due to rescue of the decreased mitochondrial membrane potential and not the fragmented mitochondria phenotype.

## Depletion of ESCRT components increases autophagic flux in *fzo-1 (tm1133)* animals

Previous studies have shown that in *C. elegans*, the depletion of ESCRT components leads to the induction of autophagy [39,40]. We confirmed this in wild-type animals (S2B Fig) and tested whether ESCRT depletion also induces autophagy in *fzo-1(tm1133)* animals. First, we determined the basal level of autophagy in *fzo-1(tm1133)* animals using three different assays that utilize the reporters P_lgg-1 GABARAP_*gfp::lgg-1* and P_sqst-1 p62_*sqst-1::gfp*, which are widely used to monitor autophagy in *C. elegans* [40,48–52]. Specifically, we determined the number of GFP::LGG-1^GABARAP foci in hypodermal seam cells of animals of the fourth larval stage (L4 larvae) and found that the average number of GFP::LGG-1^GABARAP foci increases from ~4 on average in wild-type animals (+/+) to ~23 on average in *fzo-1(tm1133)* animals (Fig 3A and 3B). As a positive control, we used RNAi against the gene *let-363*^mTOR, which induces autophagy when knocked-down [19]. As expected, *let-363(RNAi)* animals show an increase in the number of GFP::LGG-1^GABARAP foci in hypodermal seam cells (~15 on average) (Fig 3A and 3B). To determine whether the increase in the number of GFP::LGG-1^GABARAP foci is caused by a block in autophagy, we analyzed the expression of the reporter P_sqst-1 p62_*sqst-1::gfp*. (The accumulation of SQST-1^P62::GFP is indicative of defective autophagic clearance [51].) Whereas embryos homozygous for a lf mutation of *unc-51*^ULK, *e369*, a gene required for autophagy [26], show strong accumulation of SQST-1^P62::GFP, we found that *fzo-1(tm1133)* embryos do not accumulate SQST-1^P62::GFP (Fig 3C). To further verify an increase in autophagic flux in *fzo-1 (tm1133)* animals, we used an immunoblotting assay based on the cleavage of the GFP::LGG-1^GABARAP fusion protein (in autolysosomes) to generate a 'free GFP' fragment, referred to as 'cleaved GFP' [50,53,54]. As shown in Fig 3D, compared to wild type, *fzo-1(tm1133)* mutants exhibit a ~2.7-fold increase on average in the level of cleaved GFP. This confirms that autophagic flux is increased in animals lacking *fzo-1*^MFN.

To test whether depletion of ESCRT components can further increase autophagy in *fzo-1 (tm1133)* animals, we knocked-down *vps-4*^VPS4, *vps-20*^CHMP6, *hgrs-1*^HGS or *vps-37*^VPS37 in *fzo-*

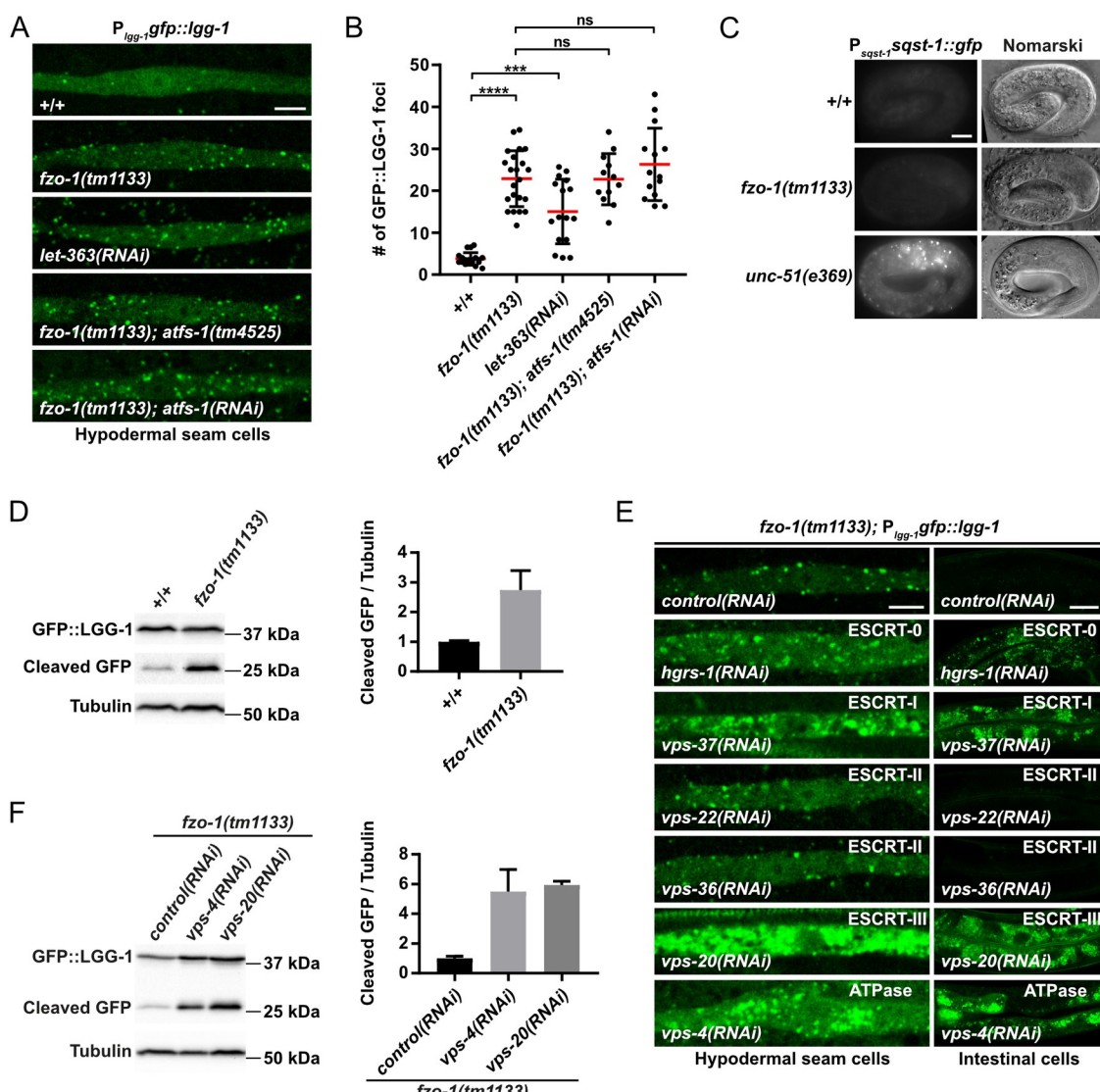

**Fig 3. Autophagy is induced independently of ATFS-1^ATF4,5 in *fzo-1(tm1133)* animals and further increased after ESCRT depletion.** (A) P_lgg-1 gfp::lgg-1 expression in hypodermal seam cells of wild type (+/+), *fzo-1(tm1133)* or *fzo-1(tm1133); atfs-1 (tm4525)* L4 larvae. For RNAi against *let-363* and *atfs-1*, L4 larvae were subjected to the respective RNAi and the F1 generation was imaged. Scale bar: 5 μm. (B) Quantification of GFP::LGG-1 foci in hypodermal seam cells from panel A. Each dot represents the average amount of GFP::LGG-1 foci counted from 2–5 seam cells in one animal. n≥12 for each genotype; values indicate means ± SD; ns: not significant, ***P<0.001, ****P<0.0001 using Kruskal-Wallis test with Dunn's multiple comparison to wild type (+/+) or *fzo-1(tm1133)*, respectively. (C) Nomarski and fluorescent images of the P_sqst-1 sqst-1::gfp translational reporter in embryos of wild type (+/+) or *fzo-1(tm1133)*. As a positive control for a block in autophagy, *unc-51(e369)* was used. Representative images of >60 embryos are shown. Scale bar: 10 μm. (D) Western blot analysis of cleaved GFP levels in wild type (+/+) or *fzo-1(tm1133)* using anti-GFP antibodies. Quantification of three independent experiments is shown. Values indicate means ± SD. (E) P_lgg-1 gfp::lgg-1 expression of *fzo-1(tm1133)* L4 larvae in hypodermal seam cells and intestinal cells upon *control(RNAi)*, *vps-4(RNAi)*, *vps-20(RNAi)*, *vps-22(RNAi)*, *hgrs-1(RNAi)*, *vps-36(RNAi)* or *vps-37(RNAi)*. Representative images of >80 animals from four independent biological replicates are shown. Scale bar hypodermal seam cells: 5 μm. Scale bar intestinal cells: 20 μm. (F) Western blot analysis of cleaved GFP levels in *fzo-1(tm1133)* upon *control(RNAi)*, *vps-4(RNAi)* or *vps-20(RNAi)* using anti-GFP antibodies. Quantification of four independent experiments is shown. Values indicate means ± SD.

*1(tm1133)* animals and analyzed GFP::LGG-1^GABARAP foci using the P_lgg-1 GABARAP gfp::lgg-1 reporter. We found that RNAi knock-down of each of these four genes in *fzo-1(tm1133)* animals causes a dramatic increase in the accumulation of GFP::LGG-1^GABARAP foci in

hypodermal seam cells as well as intestinal cells (Fig 3E). Furthermore, compared to *control (RNAi)*-treated animals, we found increased levels of cleaved GFP in *fzo-1(tm1133)* animals treated with *vps-4(RNAi)* (~5.5-fold) or *vps-20(RNAi)* (~5.9-fold) (Fig 3F). However, RNAi against the ESCRT-II components *vps-22*$^{SNF8}$ or *vps-36*$^{VPS36}$ (which fail to suppress *fzo-1 (tm1133)*-induced UPR$^{mt}$ when knocked-down (Fig 1A–1D)) has no effect on the formation of GFP::LGG-1$^{GABARAP}$ foci in hypodermal seam cells or intestinal cells (Fig 3E), probably due to an inefficient knock-down. In summary, our findings demonstrate that the depletion of components of ESCRT-0, -I, -III or the VPS-4 ATPase increases autophagic flux in *fzo-1 (tm1133)* animals.

## Induction of autophagy suppresses *fzo-1(tm1133)*-induced UPR$^{mt}$

To determine whether increasing autophagy through means other than knock-down of ESCRT components also suppresses *fzo-1(tm1133)*-induced UPR$^{mt}$, we knocked-down *let-363*$^{mTOR}$ by RNAi and examined the expression of P$_{hsp-6\ mtHSP70}$*gfp (bcSi9)* and P$_{hsp-60\ HSP60}$*gfp (zcIs9)* in *fzo-1(tm1133)* animals. We found that compared to controls, the expression of both reporters is significantly suppressed upon *let-363(RNAi)* in *fzo-1(tm1133)* animals (Fig 1A–1D). Specifically, on average, the expression of P$_{hsp-6\ mtHSP70}$*gfp* is suppressed by 40% and that of P$_{hsp-60\ HSP60}$*gfp* by 45%, which is comparable to the level of suppression observed upon RNAi knock-down of either *atfs-1*$^{ATF4,5}$ or *vps-4*$^{VPS4}$. As shown for the depletion of ESCRT components, mitochondrial morphology upon *let-363(RNAi)* was found not to be altered in *fzo-1(tm1133)* or wild-type animals (Fig 2C, 2E and 2G and S3A and S3B Fig).

To obtain further evidence that induction of autophagy leads to suppression of *fzo-1 (tm1133)*-induced UPR$^{mt}$, we searched for additional genes with a regulatory role in autophagy in our dataset of 299 suppressors. We found 17 additional genes that were previously identified in a genome-wide RNAi screen for regulators of autophagy in *C. elegans* [40] (Fig 4A). Moreover, we used a database of autophagy-related genes and their orthologs (http://www.tanpaku.org/autophagy/index.html) [55], results from two screens for regulators of autophagy in mammals [56,57], three interaction databases (wormbase.org, genemania.org and string-db.org) followed by literature searches and identified 13 additional genes in our dataset that potentially induce autophagy upon knock-down (Fig 4A) [58–74]. Therefore, including the three genes encoding components of the ESCRT (*vps-4*$^{VPS4}$, *vps-20*$^{CHMP6}$, *vps-37*$^{VPS37}$), 33 of the 299 suppressors have previously been shown to induce autophagy when knocked-down.

Finally, we knocked-down all 299 suppressors in an otherwise wild-type background and tested for an increase in autophagy. Using this approach, we found that 126 genes encode negative regulators of autophagy (16 of which were among the 33 genes identified through our literature search; indicated by $^{\S}$ in Fig 4A), since they result in the accumulation of GFP::LGG-1$^{GABARAP}$ foci in hypodermal seam cells and/or intestinal cells of larvae but not in the accumulation of SQST-1$^{p62}$::GFP in embryos when knocked-down (S1 Table). Adding the 17 genes that we identified through literature searches, which were not found in this 'autophagy' screen (Fig 4A), we, in total, found 143 out of 299 suppressors (~48%) of *fzo-1(tm1133)*-induced UPR$^{mt}$ to negatively regulate autophagy.

To confirm that the additionally identified genes enhance autophagy also in the *fzo-1 (tm1133)* background, we knocked-down six of them (*cogc-2*$^{COG2}$, *cogc-4*$^{COG4}$, *hars-1*$^{HARS}$, *rpt-3*$^{PSMC4}$, *smgl-1*$^{NBAS}$ and *ins-7*) and tested them for increased autophagic flux in *fzo-1 (tm1133)* animals. We found that the knock-down of each gene causes an increase in autophagic flux in *fzo-1(tm1133)* animals, most prominently in the intestine (Fig 4B). We also determined the level of cleaved GFP in these animals and found that, compared to *fzo-1(tm1133)*

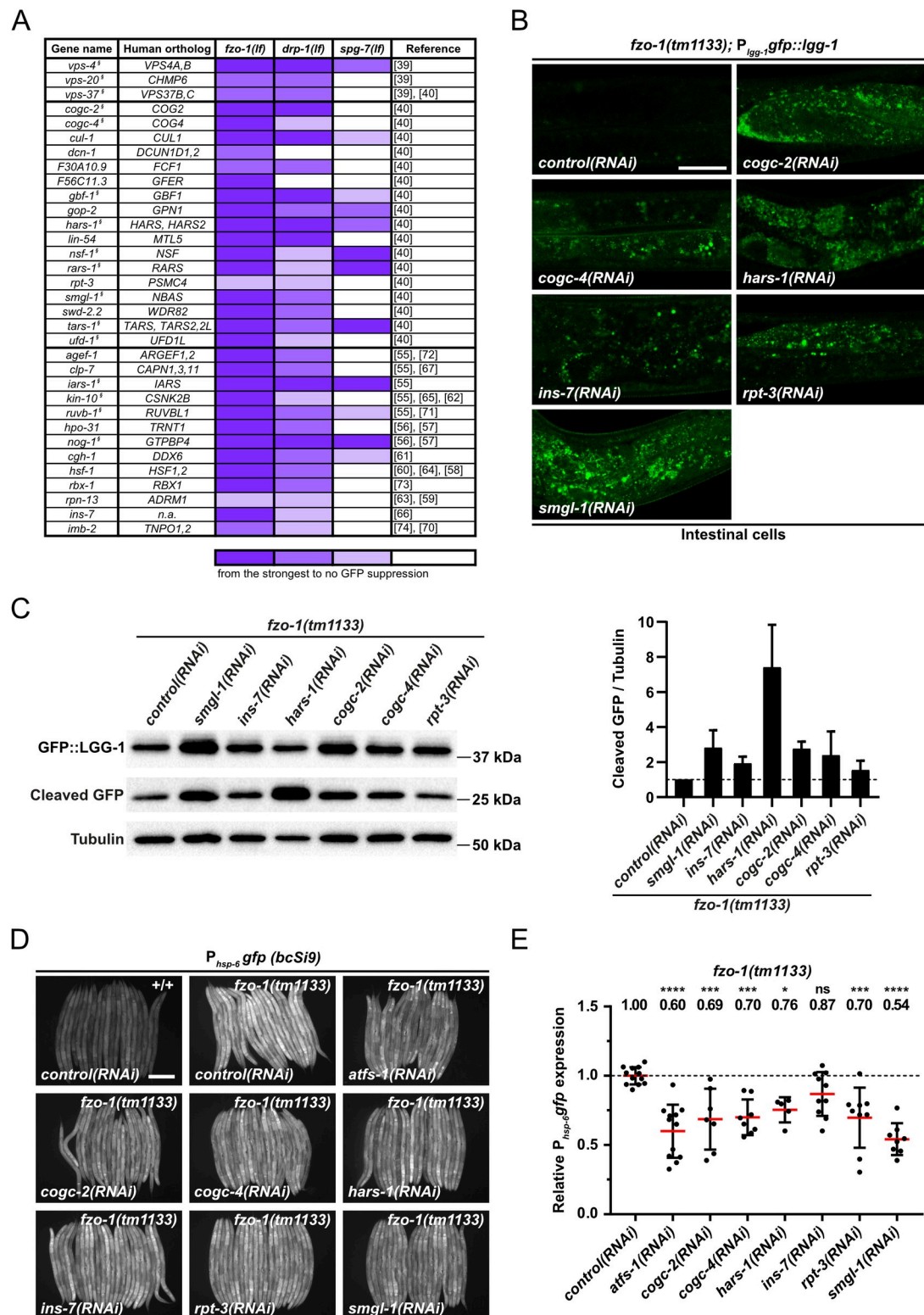

**Fig 4. Additional candidates identified by RNAi screen that suppress *fzo-1(tm1133)-* and *drp-1(tm1108)-*induced UPR**mt **through activation of autophagy. (A)** List of candidate genes identified in the primary screen with *fzo-1(tm1133);* P*hsp-6gfp (zcIs13)* by RNAi. L4 larvae were subjected to the respective RNAi and the F1 generation was imaged. Candidate genes were screened three times in technical duplicates with the same reporter in two different mutant backgrounds: *drp-1(tm1108)* and *spg-7(ad2249)*. Fluorescence intensity was scored and classified from very strong suppression to weak suppression (gradual violet

coloring) or no suppression (white). § indicates genes that, upon knock-down in our experiments, showed accumulation of GFP::LGG-1 dots in hypodermal seam cells or intestinal cells. **(B)** P$_{lgg-1}$*gfp::lgg-1* expression of *fzo-1(tm1133)* L4 larvae in intestinal cells upon *control(RNAi)*, *cogc-2(RNAi)*, *cogc-4(RNAi)*, *hars-1(RNAi)*, *ins-7(RNAi)*, *rpt-3(RNAi)* or *smgl-1(RNAi)*. Representative images of >60 animals from four independent biological replicates are shown. Scale bar: 20 μm. **(C)** Western blot analysis of cleaved GFP levels in *fzo-1(tm1133)* upon *control(RNAi)*, *smgl-1(RNAi)*, *ins-7(RNAi)*, *hars-1(RNAi)*, *cogc-2(RNAi)*, *cogc-4(RNAi)* or *rpt-3(RNAi)* using anti-GFP antibodies. Quantification of three independent experiments is shown. Values indicate means ± SD. **(D)** Fluorescence images of L4 larvae expressing P$_{hsp-6}$*gfp (bcSi9)* in wild type (+/+) or *fzo-1(tm1133)*. L4 larvae were subjected to *control(RNAi)*, *atfs-1(RNAi)*, *cogc-2(RNAi)*, *cogc-4(RNAi)*, *hars-1(RNAi)*, *ins-7(RNAi)*, *rpt-3(RNAi)* or *smgl-1(RNAi)* and the F1 generation was imaged. Scale bar: 200 μm. **(E)** Quantifications of fluorescence images from panel D. After subtracting the mean fluorescence intensity of wild type (+/+) on *control(RNAi)*, the values were normalized to *fzo-1(tm1133)* on *control (RNAi)*. Each dot represents the quantification of fluorescence intensity of 15–20 L4 larvae. Values indicate means ± SD of at least 3 independent experiments in duplicates. ns: not significant, *$P$<0.05, ***$P$<0.001, ****$P$<0.0001 using one-way ANOVA with Dunnett's multiple comparison test to *control(RNAi)*.

animals on *control(RNAi)*, the level is increased ranging from ~1.5-fold upon *rpt-3(RNAi)* to ~7.4-fold upon *hars-1(RNAi)* (Fig 4C). Using the single-copy P$_{hsp-6\ mtHSP70}$*gfp* transgene *bcSi9*, we confirmed that the knock-down of *cogc-2*[COG2], *cogc-4*[COG4], *hars-1*[HARS], *rpt-3*[PSMC4], *smgl-1*[NBAS] or *ins-7* suppresses *fzo-1(tm1133)*-induced UPR[mt] (Fig 4D and 4E). Therefore, we propose that it is the increase in autophagic flux that suppresses *fzo-1(tm1133)*-induced UPR[mt].

Since *let-363*[mTOR] as well as some of the additionally identified candidates (such as *hars-1*[HARS], *rars-1*[RARS], *tars-1*[TARS] or *iars-1*[IARS]) have roles in translation [19], we tested the effects of the depletion of *let-363*[mTOR] or *hars-1*[HARS] on P$_{ges-1\ GES2}$*gfp* expression in order to exclude that their depletion simply attenuates synthesis of GFP protein. We found that *let-363(RNAi)* or *hars-1(RNAi)* leads to suppression of P$_{ges-1\ GES2}$*gfp* expression by 39% or 25%, respectively (Fig 2A and 2B). However, we found that depletion of *let-363*[mTOR] or *hars-1*[HARS] also has a beneficial effect on mitochondrial membrane potential in *fzo-1(tm1133)* mutants since TMRE fluorescence intensity per mitochondrial area is increased by 14% or 31%, respectively while having the opposite effect in wild-type animals, in which it is decreased by 40% or 47%, respectively (Fig 2E–2H). This suggests that the suppression of *fzo-1(tm1133)*-induced UPR[mt] upon depletion of *let-363*[mTOR] or *hars-1*[HARS] is the result of a combination of an increase in mitochondrial membrane potential and the attenuation of cytosolic translation.

## The induction of autophagy is not *per se* beneficial for organismal fitness

Since mitochondrial membrane potential is increased in *fzo-1(tm1133)* animals upon induction of autophagy, we tested whether this has a beneficial effect at the organismal level. Using the 'thrashing' assay [75,76], we tested whether the motility of *fzo-1(tm1133)* animals is improved. As previously shown [77], thrashing rates are decreased in *fzo-1(tm1133)* mutants when compared to wild type (S5A Fig). We found that thrashing rates do not change upon *vps-4(RNAi)* or *vps-20(RNAi)* in either *fzo-1(tm1133)* or wild-type animals (S5B and S5C Fig). Therefore, increasing autophagic flux does not *per se* have beneficial effects on organismal fitness. In contrast, we found that thrashing rates are significantly increased upon *let-363(RNAi)* or *hars-1(RNAi)* in both *fzo-1(tm1133)* and wild-type animals (S5B and S5C Fig). Thus, the induction of autophagy can lead to increased motility under certain circumstances, but this effect may be covered upon depletion of ESCRT.

## Depletion of ESCRT components in *fzo-1(tm1133)* animals with a block in autophagy results in embryonic lethality

To test the hypothesis that increased autophagic flux is necessary for the suppression of *fzo-1(tm1133)*-induced UPR[mt] in ESCRT-depleted animals, we generated a *fzo-1(tm1133); unc-51(e369)* double mutant in the P$_{hsp-6\ mtHSP70}$*gfp (bcSi9)* reporter background and subjected it to

RNAi against either *vps-4*[VPS4] or *vps-20*[CHMP6]. However, we found that either RNAi treatment results in progeny that undergoes embryonic arrest. To circumvent this problem, we subjected *fzo-1(tm1133)* mutants to double-RNAi against *unc-51*[ULK] and *ESCRT* but failed to detect suppression of UPR[mt] upon *ESCRT(RNAi)* diluted with *control(RNAi)* (S6A Fig). Next, we depleted ESCRT components by RNAi starting from the second larval stage (L2) (rather than in the parental generation and throughout development) and examined reporter expression once the animals had reached the fourth larval stage (L4). Interestingly, we found that subjecting *fzo-1 (tm1133)* L2 larvae to *vps-4(RNAi)* or *vps-20(RNAi)* does not increase autophagic flux and fails to suppress UPR[mt], while *atfs-1(RNAi)* is able to suppress UPR[mt] under these conditions (S6B and S6C Fig). We repeated this experiment in the background of an RNAi-sensitizing mutation, *rrf-3(pk1426)*, but again were unable to detect suppression of the P$_{hsp-6\ mtHSP70}$*gfp* (*bcSi9*) reporter upon *ESCRT(RNAi)* while *atfs-1(RNAi)* suppressed (S6D Fig). Based on these results, we conclude that *ESCRT(RNAi)* does not directly act on ATFS-1[ATF4,5] to suppress UPR[mt]. Instead, we propose that it affects UPR[mt] indirectly through the induction of autophagy.

## Blocking mitophagy does not prevent suppression in *fzo-1(tm1133)* animals of UPR[mt] by ESCRT depletion

Since we were unable to test whether blocking autophagy blocks the suppression of *fzo-1 (tm1133)*-induced UPR[mt] by depletion of ESCRT components, we tested the role of *pdr-1*[Parkin]- and *fndc-1*[FUNDC1,2]-dependent mitophagy in this context [78,79]. First, we used *fzo-1(tm1133); pdr-1(lg103)* double mutants, carrying the P$_{hsp-6\ mtHSP70}$*gfp* (*bcSi9*) reporter, to test whether *pdr-1*[Parkin]-dependent mitophagy is required for ESCRT-dependent suppression of *fzo-1(tm1133)*-induced UPR[mt]. We found that knock-down of *vps-4*[VPS4], *vps-20*[CHMP6] or *hgrs-1*[HGS] still suppresses *fzo-1(tm1133)*-induced UPR[mt] in the *pdr-1(lg103)* background (Fig 5A and 5B). Furthermore, compared to the level of suppression in *fzo-1(tm1133)* animals alone, the level of UPR[mt] suppression in *fzo-1(tm1133); pdr-1(lg103)* animals is similar upon *vps-4(RNAi)* or *vps-20(RNAi)* and even higher upon *hgrs-1(RNAi)* (Figs 1A, 1C, 5A and 5B). Second, we tested whether depletion of ESCRT components suppresses UPR[mt] in *fzo-1(tm1133) fndc-1(rny14)* double mutants and found that it does so to a similar extent (Fig 5C and 5D). Therefore, *pdr-1*[Parkin]- and *fndc-1*[FUNDC1,2]-dependent mitophagy are not required for the suppression of *fzo-1 (tm1133)*-induced UPR[mt] upon ESCRT depletion.

## Blocking autophagy in the absence of mitochondrial stress induces UPR[mt], but neither blocking nor inducing UPR[mt] affects autophagy

Increasing autophagic flux suppresses *fzo-1(tm1133)*-induced UPR[mt]. To test whether decreasing autophagic flux, conversely, induces UPR[mt], we analyzed *unc-51(e369)* animals (in which autophagy is blocked) and found that compared to wild-type animals, the P$_{hsp-6\ mtHSP70}$*gfp* reporter is induced by 41% on average (Fig 5E and 5F). To determine whether the P$_{hsp-6\ mtHSP70}$*gfp* reporter is also induced under conditions where UPR[mt] is already activated, we analyzed *fzo-1(tm1133); unc-51(e369)* double mutant animals. We found that, in the *fzo-1 (tm1133)* background, the loss of *unc-51*[ULK] does not result in a significant increase in the expression of P$_{hsp-6\ mtHSP70}$*gfp* (Fig 5E and 5F). Thus, blocking autophagy induces UPR[mt] in the absence of mitochondrial stress but not under conditions where UPR[mt] is already activated.

Next, we analyzed whether blocking or inducing UPR[mt] affects autophagy. Therefore, we analyzed autophagy in animals homozygous for either the *atfs-1*[ATF4,5] lf mutation *tm4525* or the *atfs-1*[ATF4,5] gain-of-function (gf) mutation *et15gf* [11,80]. *atfs-1(tm4525)* has been shown to suppress the expression of the P$_{hsp-6\ mtHSP70}$*gfp* and P$_{hsp-60\ HSP60}$*gfp* reporters upon *spg-7*

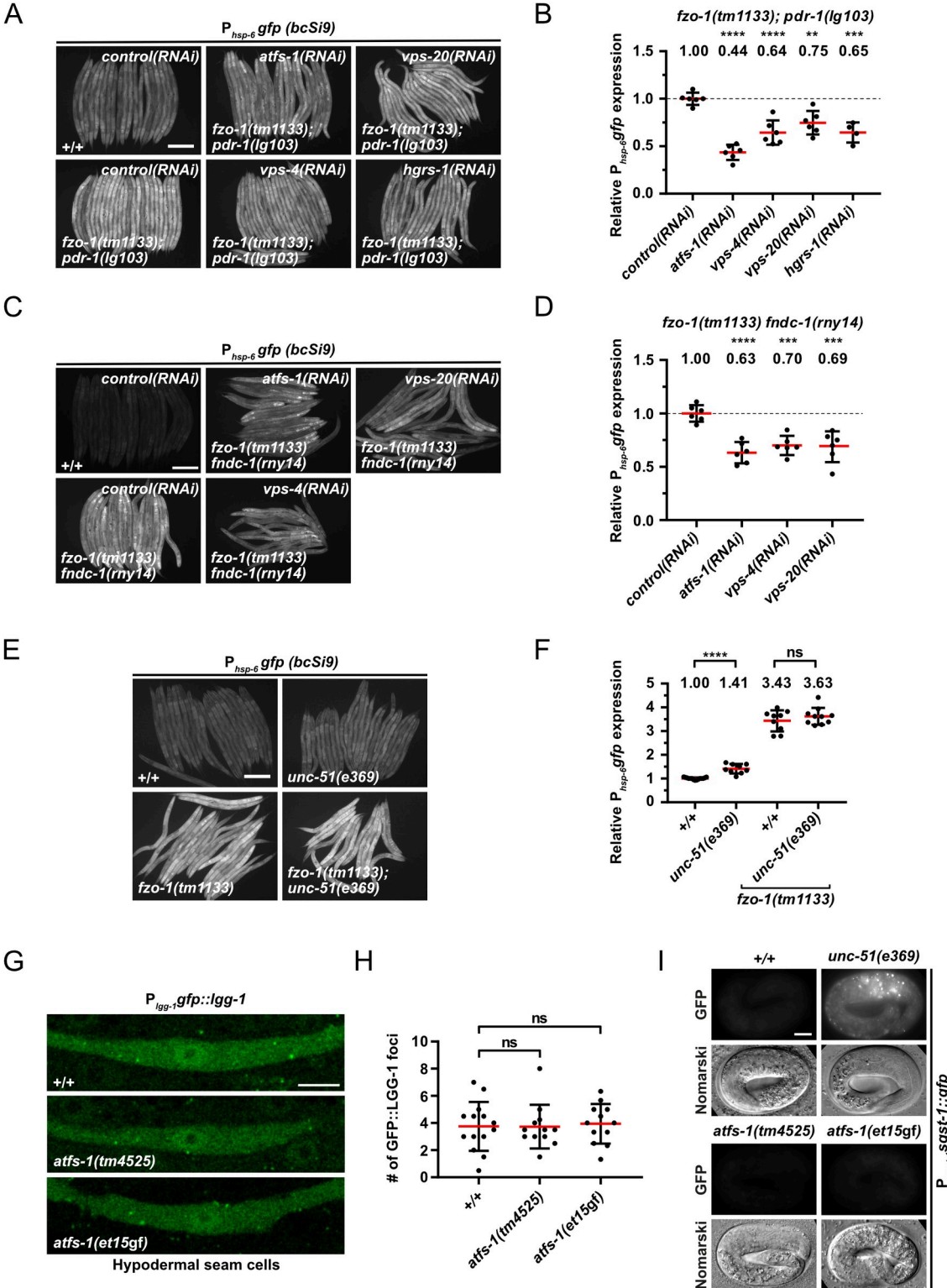

**Fig 5. Functional interactions between mitophagy, autophagy and UPR^mt.** (A) L4 larvae of *fzo-1(tm1133); pdr-1(lg103)* expressing P*hsp-6*gfp *(bcSi9)* were subjected to *control(RNAi)*, *atfs-1(RNAi)*, *vps-4(RNAi)*, *vps-20(RNAi)* or *hgrs-1(RNAi)* and the F1 generation was imaged. Scale bar: 200 μm. **(B)** Quantifications of fluorescence images from panel A. After subtracting the mean fluorescence intensity of wild type (+/+) on *control(RNAi)*, the values were normalized to *fzo-1(tm1133); pdr-1(lg103)* on *control(RNAi)*. Each dot represents the quantification of fluorescence intensity of 15–20 L4 larvae. Values indicate means ± SD of 3 independent experiments in

duplicates. **$P<0.01$, ***$P<0.001$, ****$P<0.0001$ using one-way ANOVA with Dunnett's multiple comparison test to *control(RNAi)*. **(C)** L4 larvae of *fzo-1(tm1133) fndc-1(rny14)* expressing P*hsp-6gfp (bcSi9)* were subjected to *control(RNAi)*, *atfs-1(RNAi)*, *vps-4(RNAi)* or *vps-20(RNAi)* and the F1 generation was imaged. Scale bar: 200 μm. **(D)** Quantifications of fluorescence images from panel C. After subtracting the mean fluorescence intensity of wild type (+/+) on *control(RNAi)*, the values were normalized to *fzo-1(tm1133) fndc-1 (rny-14)* on *control(RNAi)*. Each dot represents the quantification of fluorescence intensity of 15–20 L4 larvae. Values indicate means ± SD of 3 independent experiments in duplicates. ***$P<0.001$, ****$P<0.0001$ using one-way ANOVA with Dunnett's multiple comparison test to *control(RNAi)*. **(E)** Fluorescence images of L4 larvae expressing P*hsp-6gfp (bcSi9)* in wild type (+/+), *unc-51(e369)*, *fzo-1(tm1133)* or *fzo-1(tm1133); unc-51(e369)*. Scale bar: 200 μm. **(F)** Quantifications of fluorescence images from panel E. Each dot represents the quantification of fluorescence intensity of 15–20 L4 larvae. Values indicate means ± SD of at least 4 independent experiments in duplicates. ns: not significant, ****$P<0.0001$ using two-tailed t-test. **(G)** P*lgg-1gfp::lgg-1* expression in hypodermal seam cells of wild type (+/+), *atfs-1(tm4525)* or *atfs-1(et15gf)* L4 larvae. Scale bar: 5 μm. **(H)** Quantification of GFP::LGG-1 foci in hypodermal seam cells from panel G. Each dot represents the average amount of GFP::LGG-1 foci counted from 2–5 seam cells in one animal. n≥12 for each genotype; values indicate means ± SD; ns: not significant using one-way ANOVA with Dunnett's multiple comparison test to wild type (+/+). **(I)** Nomarski and fluorescent images of the P*sqst-1sqst-1::gfp* translational reporter in embryos of wild type (+/+), *atfs-1(tm4525)* or *atfs-1(et15gf)* animals. As a positive control for a block in autophagy, *unc-51(e369)* was used. Representative images of >60 embryos are shown. Scale bar: 10 μm.

*(RNAi)* and of the endogenous *hsp-6*[mtHSP70] and *hsp-60*[HSP60] loci upon *cco-1(RNAi)* [11,81]. Conversely, *atfs-1(et15gf)* has been shown to constitutively activate UPR[mt] [80]. We found that compared to wild-type animals, hypodermal seam cells of *atfs-1(tm4525)* or *atfs-1(et15gf)* animals show no significant changes in the number of GFP::LGG-1[GABARAP] foci (Fig 5G and 5H). In addition, *atfs-1(tm4525)* or *atfs-1(et15gf)* embryos do not accumulate SQST-1[P62]::GFP foci (Fig 5I). Since it has previously been reported that mitochondrial stress induces autophagy in an *atfs-1*[ATF4,5]-dependent manner [40], we also tested whether the loss of *atfs-1*[ATF4,5] suppresses autophagy in *fzo-1(tm1133)* animals. We found that the number of GFP::LGG-1[GABARAP] foci remains unchanged both in *fzo-1(tm1133)* animals upon *atfs-1(RNAi)* as well as *fzo-1(tm1133); atfs-1(tm4525)* double mutants (Fig 3A and 3B), demonstrating that the induction of autophagy in *fzo-1(tm1133)* mutants is ATFS-1[ATF4,5]-independent. Finally, we tested whether increasing UPR[mt] in *fzo-1(tm1133)* mutants by introducing *atfs-1(et15gf)* affects autophagic flux. However, we found that *fzo-1(tm1133); atfs-1(et15gf)* double mutants are not viable. Therefore, blocking or inducing UPR[mt] by manipulating ATFS-1[ATF4,5] activity does not affect autophagic flux in wild type and blocking UPR[mt] does not affect autophagy in *fzo-1(tm1133)* animals.

## The induction of autophagy suppresses UPR[mt] induced by a block in mitochondrial dynamics but not by the loss of *spg-7*[AFG3L2]

To determine whether the suppression of UPR[mt] by increased autophagic flux is specific to *fzo-1(tm1133)*-induced UPR[mt], we tested all 143 suppressors of *fzo-1(tm1133)*-induced UPR[mt] with a role in autophagy for their ability to suppress *drp-1(tm1108)*- or *spg-7(ad2249)*-induced UPR[mt] using the multi-copy P*hsp-6* mtHSP70*gfp* transgene *zcIs13*. As shown in Fig 4A and S1 Table, we found that the knock-down of 138 of the genes (~97%) also suppresses *drp-1 (tm1108)*-induced UPR[mt]. In contrast, the knock-down of 90 of the genes (~63%) suppresses *spg-7(ad2249)*-induced UPR[mt]. Among these 90 genes, 41 belong to the GO categories 'Translation' or 'Ribosome Biogenesis'. Hence, their depletion may interfere with synthesis of GFP.

Interestingly, we found that knock-down of *vps-4*[VPS4] but not *vps-20*[CHMP6] or *vps-37*[VPS37] also suppresses *spg-7(ad2249)*-induced UPR[mt] (Fig 4A). Therefore, we tested whether the knock-down of *vps-4*[VPS4] or *vps-20*[CHMP6] leads to increased autophagic flux in *spg-7(ad2249)* animals. We first analyzed the basal level of autophagy in *spg-7(ad2249)* animals using the P*lgg-1* GABARAP*gfp::lgg-1* reporter and found that compared to wild type, the number of GFP::LGG-1[GABARAP] foci is increased 2-fold (from ~4 on average in wild-type animals to ~8 on average in *spg-7(ad2249)* animals) (S7A and S7B Fig). To determine whether this increase in autophagosomes is due to a block in autophagy, we analyzed the accumulation of SQST-1[P62]::GFP using

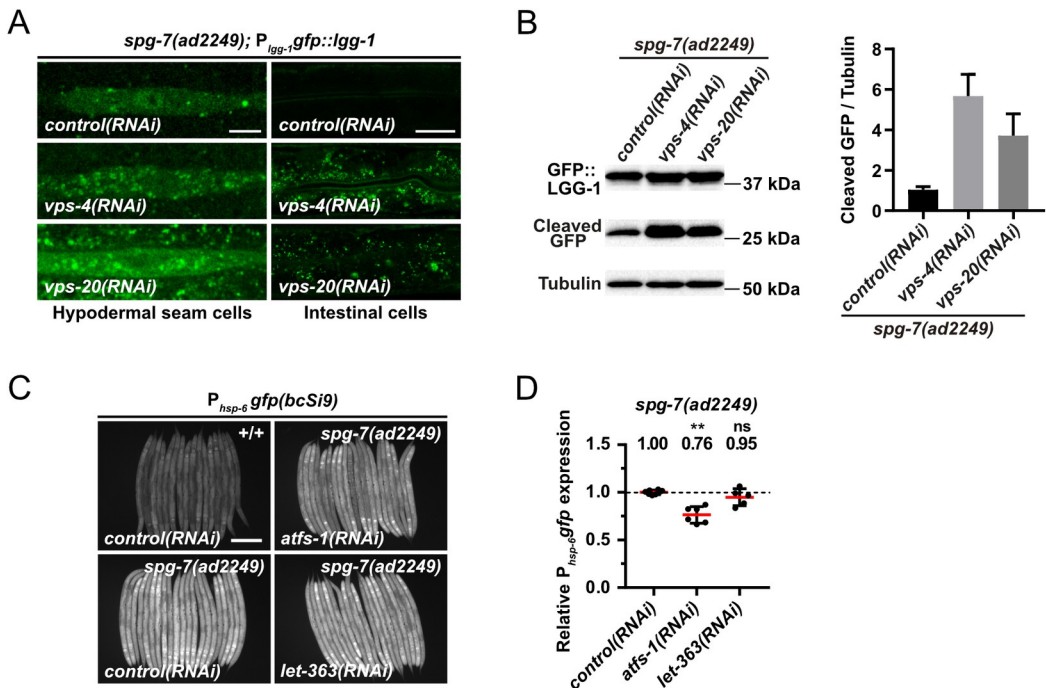

**Fig 6. Induction of autophagy is not sufficient to suppress *spg-7(ad2249)*-induced UPR^mt.** **(A)** P_*lgg-1*gfp::lgg-1 expression of *spg-7(ad2249)* L4 larvae in hypodermal seam cells and intestinal cells upon *control(RNAi)*, *vps-4(RNAi)* or *vps-20(RNAi)*. Representative images of >80 animals from three independent biological replicates are shown. Scale bar hypodermal seam cells: 5 μm. Scale bar intestinal cells: 20 μm. **(B)** Western blot analysis of cleaved GFP levels in *spg-7(ad2249)* upon *control (RNAi)*, *vps-4(RNAi)* or *vps-20(RNAi)* using anti-GFP antibodies. Quantification of three independent experiments is shown. Values indicate means ± SD. **(C)** Fluorescence images of L4 larvae expressing P_*hsp-6*gfp *(bcSi9)* in wild type (+/+) or *spg-7(ad2249)*. L4 larvae were subjected to *control(RNAi)*, *atfs-1(RNAi)* or *let-363(RNAi)* and the F1 generation was imaged. Scale bar: 200 μm. **(D)** Quantifications of fluorescence images from panel C. After subtracting the mean fluorescence intensity of wild type (+/+) on *control(RNAi)*, the values were normalized to *spg-7(ad2249)* on *control(RNAi)*. Each dot represents the quantification of fluorescence intensity of 15–20 L4 larvae. Values indicate means ± SD of 3 independent experiments in duplicates. ns: not significant, **P<0.01 using Kruskal-Wallis test with Dunn's multiple comparison test to *control(RNAi)*.

the P_*sqst-1* p62*sqst-1*::*gfp* reporter. We did not observe SQST-1^P62::GFP accumulation in *spg-7 (ad2249)* animals, thus indicating that autophagic flux is increased in *spg-7(ad2249)* mutants (S7C Fig). Next, we tested whether *vps-4(RNAi)* or *vps-20(RNAi)* further induces autophagy in the *spg-7(ad2249)* background and found that knock-down of *vps-4*^VPS4 and also *vps-20*^CHMP6 leads to an increase in the average number of GFP::LGG-1^GABARAP foci in hypodermal seam cells and intestinal cells (Fig 6A). Confirming an increase in autophagic flux, immunoblotting of GFP::LGG-1^GABARAP in *spg-7(ad2249)* animals revealed increased levels of cleaved GFP upon *vps-4(RNAi)* or *vps-20(RNAi)* (~5.7-fold and ~3.7-fold, respectively; Fig 6B). Finally, we tested whether the loss of *let-363*^mTOR, which induces autophagy and suppresses *fzo-1 (tm1133)*-induced UPR^mt (Fig 1A–1D), can suppress *spg-7(ad2249)*-induced UPR^mt. Using the single-copy P_*hsp-6* mtHSP70*gfp* transgene *bcSi9*, we found that RNAi knock-down of *let-363*^mTOR fails to suppress *spg-7(ad2249)*-induced UPR^mt (Fig 6C and 6D). In summary, these results indicate that UPR^mt induced by the loss of *spg-7*^AFG3L2 is not suppressed by increasing autophagic flux. Based on these findings we propose that the induction of autophagy is sufficient to suppress UPR^mt induced by a block in mitochondrial dynamics but not by the loss of *spg-7*^AFG3L2.

## Defects in mitochondrial dynamics lead to changes in the levels of certain types of triacylglyerols, which can partially be reverted by induction of autophagy

To elucidate how the induction of autophagy leads to suppression of UPR^mt in *fzo-1(tm1133)* and *drp-1(tm1108)* animals, we determined potential differences in metabolism in these genetic backgrounds. Since mitochondria and autophagy are known to regulate specific aspects of lipid metabolism, we performed non-targeted lipid profiling in *fzo-1(tm1133)*, *drp-1 (tm1108)* and *spg-7(ad2249)* mutant backgrounds and compared them to wild type.

Of the 5284 lipid 'features' detected, the levels of 3819 are changed in at least one of the three pairwise comparisons (*fzo-1(tm1133)* vs. wild type, *drp-1(tm1108)* vs. wild type, *spg-7 (ad2249)* vs. wild type) (S8A Fig). Among the 3819 lipid features that are changed, 1774 are currently annotated as lipids. Interestingly, a third of the annotated lipids, whose levels were changed, are triacylglycerols (TGs). TGs are storage lipids and make up a major part of lipid droplets, which are broken down into fatty acids and subsequently oxidized in mitochondria upon energy demand [82–84]. We initially determined the total amounts of TGs in the mutant backgrounds and compared them to that of wild type. Whereas *drp-1(tm1108)* mutants show an increase in the total amount of TGs, no changes are observed in *fzo-1(tm1133)* mutants and a decrease is detected in *spg-7(ad2249)* mutants (S8B Fig). To determine whether the amounts of TG species with a specific length of acyl chains and/or number of double bonds are altered, we plotted all 659 detected TGs and subsequently marked TGs that are specifically up- (red) or downregulated (blue) in *fzo-1(tm1133)*, *drp-1(tm1108)* or *spg-7(ad2249)* animals (S8C Fig and S2 Table). Consistent with the observed decrease in the total amount of TGs, most of the individual TG species are downregulated in *spg-7(ad2249)* mutants (S8B and S8C Fig). In the *drp-1(tm1108)* background, TG species with altered levels initially showed no distinct pattern regarding length of acyl chains or degree of desaturation (S8C Fig and S2 Table). However, in the *fzo-1(tm1133)* background, these TG species can be separated into two clusters. Whereas TGs with shorter acyl chains are downregulated in *fzo-1(tm1133)* mutants, 'longer' TGs with a higher degree of unsaturation are increased (S8C Fig and S2 Table). Interestingly, when looking at the overlap between *fzo-1(tm1133)* and *drp-1(tm1108)*, we observed a similar trend regarding changes in acyl length and desaturation for *drp-1(tm1108)* as well (S8D Fig and S2 Table).

Next, we tested whether the induction of autophagy can revert the specific changes in TG pattern observed in *fzo-1(tm1133)* mutants. Therefore, we knocked-down *vps-4*^VPS4 or *cogc-2*^COG2 to induce autophagy in *fzo-1(tm1133)* and wild-type animals and again, performed lipid profiling. We used principal component analysis (PCA) in order to show how distinct or similar the lipid profiles upon *vps-4(RNAi)* or *cogc-2(RNAi)* are. Interestingly, knock-down of *vps-4*^VPS4 in either genotype was distinct from controls, which indicates major changes in the lipidome due to an efficient RNAi knock-down (Fig 7A). Moreover, we found that RNAi against *cogc-2*^COG2 has only mild effects, since the samples cluster with controls in both genotypes. This might be attributed to a weak knock-down and most probably a weak induction of autophagy.

Subsequently, we specifically analyzed the TGs in *fzo-1(tm1133)* mutants on *control(RNAi)* and, consistent with our previous results (S8C Fig (left panel) and S2 Table), detected a decrease in the levels of TGs with shorter acyl chains while levels of TGs with longer chains increase, compared to wild type on *control(RNAi)* (Fig 7B (left panel) and S2 Table). The levels of TGs that are downregulated in the *fzo-1(tm1133)* background are either unchanged or further decreased upon depletion of *vps-4*^VPS4 and the concomitant induction of autophagy (Fig 7B (middle panel) and S2 Table). In contrast, the levels of TGs that are upregulated in *fzo-1*

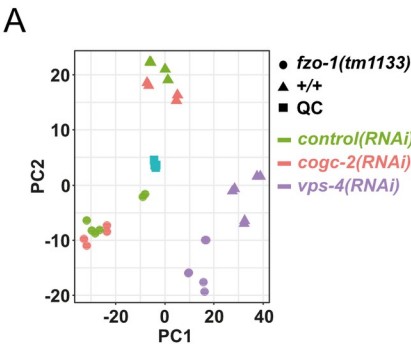

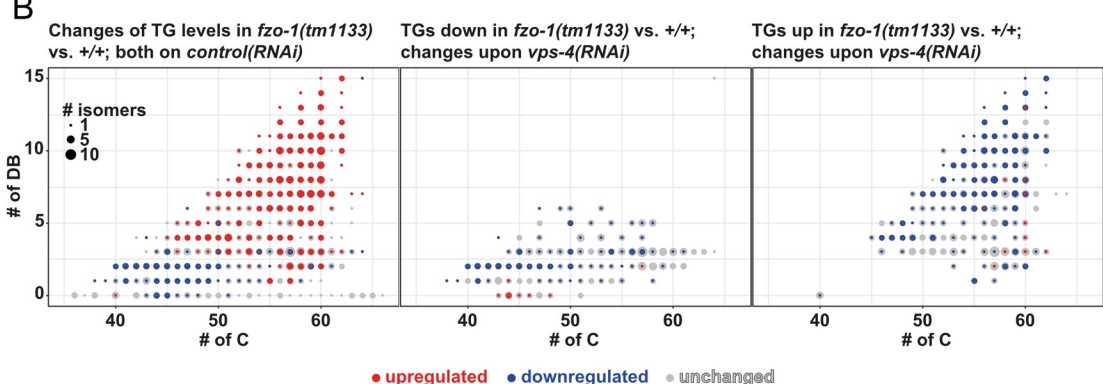

**Fig 7. Induction of autophagy upon *vps-4(RNAi)* changes the levels of specific TGs in *fzo-1(tm1133)* mutants.** **(A)** Principal component analysis (PCA) scores plot of wild-type (+/+) and *fzo-1(tm1133)* animals subjected to *control(RNAi)*, *cogc-2(RNAi)* or *vps-4(RNAi)*. Turquois squares indicate internal quality controls (QC). **(B)** Scatterplot indicating the distribution and changes in the levels of TG species in *fzo-1(tm1133)* mutants in comparison to wild type (+/+). The x-axis labels the number of carbons (# of C) and the y-axis the number of double bonds (DB) in the acyl sidechains. The size of a dot indicates the number of detected isomers for a specific sum composition. Grey dots represent all detected TGs species and blue and red dots indicate down- (blue) or upregulation (red).

*(tm1133)* animals are reduced upon induction of autophagy by knock-down of *vps-4*<sup>VPS4</sup>, although not always to the levels of wild type (Fig 7B (right panel) and S2 Table). Upon *cogc-2 (RNAi)*, we detected only minor effects on the levels of TGs in *fzo-1(tm1133)* (S9A Fig and S2 Table), which is consistent with the relatively small changes in the lipid profile as assessed by PCA (Fig 7A). However, the levels of most TGs that are decreased upon *cogc-2(RNAi)* are also decreased upon depletion of *vps-4*<sup>VPS4</sup> (S9B Fig), suggesting that the induction of autophagy caused by the two different knock-downs leads to partially overlapping changes in the levels of TGs. Taken together, we find that the levels of specific TGs are changed in a similar manner in mutants with defects in mitochondrial dynamics. Moreover, we show that some of these changes can be reverted by the induction of autophagy in *fzo-1(tm1133)* animals.

## Discussion

### Induction of autophagy increases mitochondrial membrane potential and suppresses UPR<sup>mt</sup> in *fzo-1(tm1133)* mutants

We propose that the induction of autophagy partially restores membrane potential and thereby suppresses *fzo-1(tm1133)*-induced UPR<sup>mt</sup>. Interestingly, a decrease in mitochondrial membrane potential has recently been shown to be the signal for UPR<sup>mt</sup> induction [10]. Therefore, some aspect of mitochondrial stress that leads to both decreased membrane potential and the

induction of UPR$^{mt}$ in *fzo-1(tm1133)* mutants can be rescued by the induction of autophagy in these animals. We were unable to verify our hypothesis since ESCRT-depleted *fzo-1(tm1133); unc-51(e369)* double mutants arrest during embryogenesis. This is in agreement with a study from Djeddi *et al.*, which reported that induction of autophagy is a pro-survival mechanism in ESCRT-depleted animals [39]. Moreover, our data suggests that clearance of defective and depolarized mitochondria by *pdr-1*$^{Parkin}$- or *fndc-1*$^{FUNDC1,2}$-dependent mitophagy does not play a role in the suppression of *fzo-1(tm1133)*-induced UPR$^{mt}$. In addition, we propose that the induction of autophagy may lead to increased organismal fitness, but that this effect is masked by pleiotropic effects upon knock-down of certain genes such as the *ESCRT* genes.

## Increased autophagic flux compensates for a block in mitochondrial dynamics

We provide evidence that the induction of autophagy can also compensate for a block in mitochondrial fission and, hence, for defects in mitochondrial dynamics. In contrast, induction of autophagy does not suppress *spg-7(ad2249)*-induced UPR$^{mt}$. Among the genes that suppress *spg-7(ad2249)*-induced UPR$^{mt}$ almost half have roles in translation or ribosome biogenesis, the knock-down of which may impair GFP synthesis by compromising cytosolic translation. Furthermore, we speculate that the knock-down of the remaining genes suppresses *spg-7 (ad2249)*-induced UPR$^{mt}$ through mechanisms other than the induction of autophagy. This supports the notion that UPR$^{mt}$ induced by different types of mitochondrial stress are distinct in their mechanisms of induction and also in their mechanisms of suppression. In line with this, we found that different mitochondrial stresses have different impacts on the lipidome. Although FZO-1 and DRP-1 play different roles in mitochondrial dynamics, they have similar effects on the levels of many TGs when mutated. In contrast, the levels of these TGs are distinct in *spg-7(ad2249)* animals. The role of mitochondria in the metabolism of TGs is diverse. First, mitochondria are using fatty acids released from TGs upon lipolysis for energy production. Second, lipid droplet associated mitochondria deliver building blocks and energy for the synthesis of fatty acids and TGs. Fatty acids derived from this pathway typically show lower chain length and a higher degree of saturation [85]. Since we see a decrease in TGs with shorter chain length in *fzo-1(tm1133)* mutants, it is plausible that contact sites between lipid droplets and mitochondria are affected. Consistent with this, Benador *et al.* found high levels of MFN2 in lipid droplet associated mitochondria in brown adipose tissue of mice [85]. Furthermore, Rambold *et al.* reported that altered mitochondrial morphology in mouse embryonic fibroblasts lacking either *Opa1* or *Mfn1* affects fatty acid transfer from lipid droplets to mitochondria, thereby causing heterogeneous fatty acid distribution across the mitochondrial population [86]. Therefore, we speculate that the loss of *fzo-1*$^{MFN}$ or *drp-1*$^{DRP1}$ but not *spg-7*$^{AFG3L2}$ leads to alterations in contact sites between lipid droplets and mitochondria and that these alterations lead to specific changes in metabolism.

Interestingly, we found that increasing autophagic flux in *fzo-1(tm1133)* animals reverts some of the changes in the levels of TGs. Consistent with these results, autophagy has been shown to have a role in the breakdown of TGs from lipid droplets, which ensures a constant fatty acid supply to mitochondria for β-oxidation [87], highlighting the importance of autophagy in fatty acid metabolism. More recently, autophagy has also been shown to directly affect the levels of enzymes involved in β-oxidation by causing the degradation of the co-repressor of PPARα, a master regulator of lipid metabolism [88]. Therefore, we propose that the induction of autophagy in mutants with defects in mitochondrial dynamics results in elevated breakdown of specific TGs that are used to fuel mitochondrial metabolism, thereby leading to increased mitochondrial membrane potential and suppression of UPR$^{mt}$.

## Functional interactions between autophagy and UPR<sup>mt</sup>

Protection of mitochondrial and ultimately cellular homeostasis was previously proposed to be dependent on the integration of different mitochondrial and cellular stress pathways but experimental data so far was limited [89]. The first evidence that autophagy can affect UPR$^{mt}$ was the finding by Haynes *et al.* that knock-down of *rheb-1*$^{RHEB}$, a known positive regulator of TOR [90], suppresses the P$_{hsp-60}$ $_{HSP60}$*gfp* reporter [13]. Two more recent studies reported contradictory results with respect to the effect of blocking mitophagy on UPR$^{mt}$ induction [7,91]. We demonstrate that a block in autophagy in the absence of mitochondrial stress induces UPR$^{mt}$. Blocking autophagy results in major changes in metabolism [92,93] which may, to some extent, be caused by decreased delivery of lipids into mitochondria. This could consequently lead to the activation of UPR$^{mt}$ and thereby to a metabolic shift towards glycolysis [94]. Thus, *fzo-1(tm1133)* mutants, in which UPR$^{mt}$ is already activated, are less dependent on their mitochondria with regard to energy production and this might explain why blocking autophagy in these animals does not further increase UPR$^{mt}$. Interestingly, based on our results, altering autophagy can influence UPR$^{mt}$, but changes in UPR$^{mt}$ do not affect autophagy. In contrast, Guo *et al.* reported that upon mitochondrial stress, upregulation of both UPR$^{mt}$ and autophagy is dependent on ATFS-1$^{ATF4,5}$ [40] and Nargund *et al.* showed that a small subset of autophagy related genes are upregulated via ATFS-1$^{ATF4,5}$ upon mitochondrial stress (induced by *spg-7(RNAi)*) [11]. However, we show that import of ATFS-1$^{ATF4,5}$ into the nucleus under conditions where mitochondrial stress is absent, is not sufficient to induce autophagy. Taken together, we found a previously undescribed functional connection between autophagy and UPR$^{mt}$. We propose that the two pathways do not interact directly but that the induction of autophagy leads to improved mitochondrial function by affecting lipid metabolism and ameliorating cellular homeostasis, thereby suppressing UPR$^{mt}$ in mutants with defects in mitochondrial dynamics (Fig 8).

## Genome-wide RNAi screen identifies a new autophagy network

In our dataset of 299 suppressors of *fzo-1(tm1133)*-induced UPR$^{mt}$ we found 143 genes that negatively regulate autophagy. Interestingly, 94% of these candidates (135/143) have orthologs in humans. We identified several components of the ubiquitin-proteasome system (UPS) (*rpt-3*$^{PSMC4}$, *rpn-13*$^{ADRM1}$, *ufd-1*$^{UFD1}$, *rbx-1*$^{RBX1}$, *cul-1*$^{CUL1}$) [73,95,96] and found evidence in the literature that activation of autophagy compensates for the loss of the UPS [59,63]. Additionally, we identified several genes that are involved in cell signaling, e.g. *ruvb-1*$^{RUVBL1}$, a

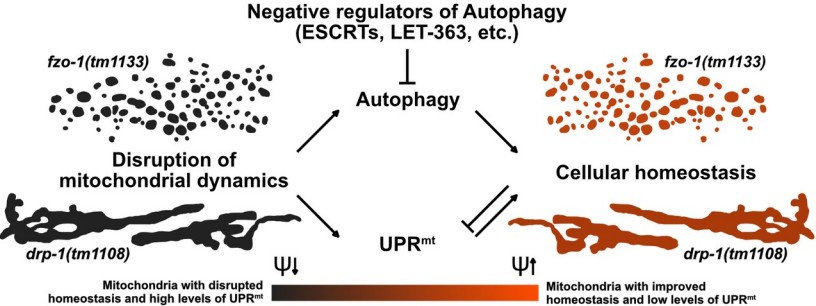

**Fig 8. Autophagy compensates for defects in mitochondrial dynamics.** The disruption of mitochondrial dynamics leads to altered mitochondrial morphology and to activation of UPR$^{mt}$ and autophagy. We propose that in animals with compromised mitochondrial dynamics, the induction of autophagy fuels mitochondrial metabolism, thereby leading to increased mitochondrial membrane potential ($\psi$) and improved cellular homeostasis, which consequently results in suppression of UPR$^{mt}$.

component of the TOR pathway in *C. elegans* that induces autophagy when knocked-down [71]. Among the genes with roles in cellular trafficking, we found *imb-2*[TNPO1,2], a regulator of the nuclear transport of DAF-16[FOXO] [70], which has been implicated in the regulation of autophagy [74]. Approximately one third of the candidates identified (44/143) are genes that regulate protein biosynthesis (S1 Table, GO categories 'Ribosome Biogenesis' and 'Translation'), which was shown to be protective against mitochondrial stress when impaired [97]. Baker and colleagues showed that knock-down of protein kinases involved in translation, such as *let-363*[mTOR], specifically suppress P$_{hsp-60\ HSP60}$*gfp (zcIs9)* expression. Based on our results, we propose that this effect could, to some extent, be due to the induction of autophagy. Taken together, we identified a broad range of cellular components and processes that all impact autophagy when deregulated, demonstrating the diverse and critical roles of autophagy in cellular homeostasis.

## Conclusions

A block in mitochondrial dynamics leads to decreased mitochondrial membrane potential and the induction of UPR[mt]. Lipid profiling indicates that a block in mitochondrial dynamics also causes an increase in the levels of certain types of TGs, which is reversed by induction of autophagy. We propose that the breakdown of these TGs through an autophagy-dependent process leads to elevated metabolic activity and that this causes an increase in mitochondrial membrane potential and the suppression of UPR[mt].

## Methods

### General *C. elegans* methods and strains

*C. elegans* strains were cultured as previously described [98]. Bristol N2 was used as the wild-type strain and the following alleles and transgenes were used: LGI: *spg-7(ad2249)* [41]; LGII: *fzo-1(tm1133)* (National BioResource Project), *rrf-3(pk1426)* [99], *fndc-1(rny14)* [78]; LGIII: *pdr-1(lg103)* [100]; LGIV: *drp-1(tm1108)* (National BioResource Project), *bcSi9* (P$_{hsp-6}$::*gfp*::*unc-54 3'UTR*) (this study), *frIs7 (nlp-29p::GFP + col-12p::DsRed)* [101]; LGV: *unc-51(e369)* [23], *atfs-1(tm4525)* (National BioResource Project), *atfs-1(et15*gf*)* [80]. Additionally, the following multi-copy integrated transgenes were used: *adIs2122(lgg-1p::GFP::lgg-1 + rol-6 (su1006))* [102], *bpIs151 (sqst-1p::sqst-1::GFP + unc-76(+))* [51], *zcIs9 (P$_{hsp-60}$::gfp::unc-54 3'UTR)* [14], *zcIs13 (P$_{hsp-6}$::gfp::unc-54 3'UTR)* [14], *zcIs18 (P$_{ges-1}$::gfp(cyt))* [103], *bcIs79 (P$_{let-858}$::gfp$^{mt}$::let-858 3'UTR + rol-6(su1006))*, *bcIs78 (P$_{myo-3}$::gfp$^{mt}$::unc-54 3'UTR + rol-6(su1006))* [46]. The strains MOC92 *bicIs10(hsp-1::tagRFP::unc-54 3'UTR)* and MOC119 *bicIs12(ttr-45p::tagRFP::ttr-45 3'UTR)* were generated in the Casanueva lab by gonadal microinjection of plasmids pMOC1 and pMOC2, respectively followed by genome integration via UV irradiation using a Stratagene UV Crosslinker (Stratalinker) [104]. The irradiation dose was 35mJ/cm$^2$ corresponding to Stratalinker power set up at 350. The single-copy integration allele *bcSi9* was generated using MosSCI [105] of the plasmid pBC1516. The strain EG8081 (*unc-119(ed3) III; oxTi177 IV*) was used for targeted insertion on LGIV [106]. The strain MD2988 (P$_{let-858}$*gfp$^{mt}$*) was generated by gonadal microinjection of the plasmid pBC938 followed by genome integration via EMS mutagenesis.

### Plasmid construction

The plasmid pBC1516 was constructed using Gibson assembly [107]. The vector pCFJ350 (a gift from Erik Jorgensen; Addgene plasmid no. 34866) [108] was digested using AvrII. The putative *hsp-6* promoter (1695bp upstream of the start codon of *hsp-6*) + 30 bp of the *hsp-6*

gene were PCR amplified from gDNA using overhang primers to pCFJ350 5'- acgtcaccggttcta-gatacTCGAGTCCATACAAGCACTC -3' and *gfp::unc-54 3'UTR* 5'- ctttactcatGGAAGACAA GAATGATCGTG -3' (lower case letters indicating overhangs). *gfp::unc-54 3'UTR* was PCR amplified from pPD95.77 using overhang primers to P*$_{hsp-6}$* 5'- cttgtcttccATGAGTAAAGGA GAAGAACTTTTC -3' and pCFJ350 5'- tagagggtaccagagctcacAAACAGTTATGTTTGGTA TATTGG -3' (lower case letters indicating overhangs).

The plasmid pBC938 was constructed using a classical cloning approach. Therefore, *gfp$^{mt}$* was amplified by PCR from pBC307 (P*$_{hs}$gfp$^{mt}$*) [109] using the following primers carrying a NheI or KpnI restriction site, respectively:

mitogfpFKpnI: 5'- GGTACCATGGCACTCCTGCAATCAC -3'
mitogfpRNheI: 5'- GCTAGCCTATTTGTATAGTTCATCCATGC -3'

The amplified fragment was then digested with KpnI and NheI and subsequently ligated into the NheI and KpnI digested backbone L3786 (P*$_{let-858}$NLS-GFP*) (L3786 was a gift from Andrew Fire (Addgene plasmid # 1593; http://n2t.net/addgene:1593; RRID:Addgene_1593)).

The plasmids pMOC1 and pMOC2 were generated by Gibson cloning, using Gibson Assembly Master Mix (New England Biolabs E2611) according to standard protocol using the vector pTagRFP-C as backbone (Evrogen). For the plasmid pMOC1 *(hsp-1p::tagRFP::unc-54 3'UTR))*, the 1.3 kb intergenic region upstream *hsp-1* was amplified and inserted at ScaI site, using the following primers:

hsp-1p fwd: 5'- GCCTCTAGAGTTACTTCGGCTCTATTACTG -3'
hsp-1p rev: 5'- tatcgcgagtTTTTACTGTAAAAAATAATTTAAAAATCAAGAAATAG -3'

The 3'UTR of *unc-54* was amplified and inserted at XhoI site using the primers:

unc54UTR RFP fwd: 5'- CTTAATTaaAGGACTCAGATCgtccaattactcttcaacatc -3'
unc54UTR RFP rev: 5'- CAGAATTCGAAGCTTGAGCttcaaaaaaatttatcagaag -3'

For the plasmid pMOC2 *(ttr-45p::tagRFP::ttr45 3'UTR)*, the 1.85 kb intergenic region upstream *ttr-45* was amplified and inserted at XbaI site, using the following primers:

ttr-45p fwd: 5'- GCCTGCAGGCGCGCCTctgaaaaaaaatcatattacaaatcag -3'
ttr-45p rev: 5'- AGATATCGCGAGTACTtgaaattttaaattttgaattttagtc -3'

The 3'UTR of *ttr-45*, contained in the following primer (lower case) was inserted at the XhoI site:

ttr-45UTR:
5'- TTaaAGGACTCAGATCaataattttgattttatgtataataaagactttatctcggGCTCAAGCTTCGAA TT -3'

## RNA-mediated interference

RNAi by feeding was performed using the Ahringer RNAi library [45]. *sorb-1(RNAi)* was used as a negative control (referred to as '*control(RNAi)*') in all RNAi experiments. For all experiments, except for the screens in *fzo-1(tm1133)*, *drp-1(tm1108)* and *spg-7(ad2249)*, RNAi clones were cultured overnight in 2 mL of LB carbenicillin (100 μg/mL) at 37˚C and 200 rpm. The RNAi cultures were adjusted to 0.5 OD and 50 μL were used to seed 30 mm RNAi plates containing 6 mM IPTG. The plates were incubated at 20˚C in the dark. 24 hours later, two L4 larvae of all wild-type strains or 16 L4 larvae of all strains carrying the *fzo-1(tm1133)* allele were inoculated onto the RNAi plates. L4 larvae of the F1 generation were collected after 4 days (wild-type strains) or 6–7 days (*fzo-1(tm1133)* mutants). *hars-1(RNAi)* was diluted 1:5 with *sorb-1(RNAi)* in all experiments. Larvae were imaged using M9 buffer with 150 mM sodium azide.

For the screens with the multi-copy *zcIs13* transgene in *fzo-1(tm1133)*, *drp-1(tm1108)* and *spg-7(ad2249)*, RNAi clones were cultured overnight in 100 μL of LB carbenicillin (100 μg/mL)

in a 96 well plate format at 37°C and 200 rpm. 10 μL of the RNAi cultures was used to seed 24 well RNAi plates containing 0.25% Lactose (w/v). The plates were incubated at 20°C in the dark. 24 hours later, 3 L4 larvae of all strains carrying the *fzo-1(tm1133)* and *spg-7(ad2249)* allele, and 2 L4 larvae of *drp-1(tm1108)* were inoculated onto the RNAi plates. The F1 generation was scored by eye for fluorescence intensity after 4–7 days.

### Image acquisition, processing and analysis

For each RNAi condition, 10–20 animals were immobilized with M9 buffer containing 150 mM sodium azide on 2% agarose pads and imaged at 100x using a Leica GFP dissecting microscope (M205 FA) and the software Leica Application Suite (3.2.0.9652).

For image analysis, we used a Fiji-implemented macro using the IJ1 Macro language to automate the intensity measurement within defined areas of 2-dimensional images. An automated threshold using the Triangle method was applied to the fluorescence microscopy image, in order to generate a binary mask (The Triangle method was selected among the 16 available auto threshold methods of ImageJ as it provided the best results.). The mask was then inverted and the Particle Analyzer of ImageJ was used to remove noise by setting a minimum size (10 pixels) for objects to be included in the mask. After manually removing any remaining unwanted objects, the mask was applied to the corresponding fluorescent microscopy image and mean fluorescent intensity was measured. The mean fluorescent intensity outside the mask was defined as the background.

Mitochondrial morphology was assessed in a strain carrying *bcIs78* and *bcIs79* using a Zeiss Axioskop 2 and MetaMorph software (Molecular Devices).

### TMRE staining and quantification

TMRE staining was performed with the F1 generation of respective RNAi treatments. L2 larvae were inoculated onto plates containing 0.1 μM TMRE (Thermo Life Sciences T669) and imaged in L4 stage using a 63x objective on Zeiss Axioskop 2 and MetaMorph software (Molecular Devices). Thereby TMRE is used in non-quenching mode and therefore suitable for quantifications and direct correlations to mitochondrial membrane potential.

The image is first converted to an 8-bit image, after which the continuous background signal is removed through background subtraction using the "rolling ball" algorithm with a ball radius of 15 pixels [110]. To remove remaining noise, two filters are applied. The first being a minimum filter with a value of 1, therefore replacing each pixel in the image with the smallest pixel value in a particular pixel's neighborhood. This is followed by a mean filter with a radius of 2, which replaces each pixel with the neighborhood mean. Next, the Tubeness plugin is run with a sigma value of 1.0, which generates a score of how tube-like each point in the image is by using the eigenvalues of the Hessian matrix to calculate the measure of "tubeness" [111]. The resulting 32-bit image is converted back to 8-bit and an automatic threshold (using the IsoData algorithm) generates a binary mask. The final step involves the removal of any particles that are smaller than 10 pixels in size for they are assumed to be noise.

Raw image files are opened in parallel to their appendant binary masks (generated by the segmentation macro) and a mask-based selection is created in the raw image. Within this selection measurements are obtained in the raw image and collected for subsequent analysis.

### Western blot analysis

Mixed-stage populations of worms were harvested, washed three times in M9 buffer, and the pellets were lysed in 2x Laemmli buffer. For analysis of the additional candidates (Fig 4) 60–80 L4 stage animals were picked for western blotting. For analysis of endogenous HSP-6, 100 L4

larvae were harvested per genotype. The protein extracts were separated by 10% SDS-PAGE and transferred to a PVDF membrane (0.45 μm pore, Merck Millipore). To detect GFP and Tubulin, we used primary anti-GFP (1:1000, Roche 11814460001) and primary anti-α-Tubulin (1:5000, Abcam ab7291) antibodies and secondary horseradish peroxidase-conjugated goat anti-mouse antibodies (BioRad #1706516). To detect endogenous HSP-6, we used anti-HSP-6 (1:10,000) as described previously [42] and secondary horseradish peroxidase-conjugated goat anti-rabbit antibodies (BioRad #1706515). Blots were developed using ECL (Amersham) or ECL Prime (Amersham) according to manufacturer's protocol and images were quantified using the ChemiDoc XRS+ System (BioRad).

## Analysis of autophagy and quantification of GFP::LGG-1 foci

L4 stage animals (except otherwise mentioned) were immobilized with M9 buffer containing 150 mM sodium azide on 2% agarose pads. Animals were imaged using a Leica TCS SP5 II confocal microscope (Leica Application Suite LAS software) with a 63x objective. GFP fluorescence was detected by excitation at 488 nm and emission at 507–518 nm. GFP::LGG-1 foci were counted in hypodermal seam cells on single images where the nucleus could clearly be seen. The amount of GFP::LGG-1 foci was counted in 2–5 seam cells per animal and the average number of GFP::LGG-1 foci per hypodermal seam cell was plotted for graphical representation and statistical analysis. SQST-1::GFP was imaged using Zeiss Axioskop 2 and MetaMorph software (Molecular Devices).

## Analysis of thrashing rate

Body bends of L4 larvae were counted as previously described [75]. Briefly, the animals were transferred from the RNAi plates onto an empty NGM plate to get rid of all bacteria and then subsequently transferred into an empty petri dish filled with M9 buffer. After letting the L4 larvae adjust for one minute, they were recorded using a Samsung Galaxy S8 attached to a Leica MS5 stereomicroscope. The videos were played back at reduced speed using VLC media player (v3.0.8) and the number of body bends was counted manually for 1 minute.

## Statistics

For experiments where two groups were compared, datasets were first tested for normality using Shapiro-Wilk normality test. If all samples of one dataset were found to be normally distributed, we conducted an unpaired two-tailed t-test. If samples were found to have non-equal variance, we conducted an unpaired tow-tailed t-test with Welch's correction. For experiments where more than two groups were compared, datasets were first tested for normal distribution using Shapiro-Wilk normality test and then tested for equal variance using Brown-Forsythe test. If samples of one dataset were found to be normally distributed and to have equal variance, one-way ANOVA with Dunnett's post hoc test was used to test for statistical significance with multiple comparisons to controls. If the dataset was not found to have normal distribution and/or have equal variance, Kruskal-Wallis test with Dunn's post hoc test for multiple comparisons to controls was used.

## Lipid profiling using UPLC-UHR-ToF-MS

RNAi in lipidomic experiments was performed using *OP50(xu363)*, which is compatible for dsRNA production and delivery [112]. The L4440 plasmids containing the coding sequence of *sorb-1*, *cogc-2* or *vps-4* were purified from HT115 bacteria of the Ahringer library [45] using Qiagen Plasmid Mini Kit (Cat. No. 12125) and subsequently transformed into chemically

competent *OP50(xu363)*. Single clones were picked, sequenced and glycerol stocks were made for subsequent experiments. Bacterial clones were grown as described in section 'RNA-mediated interference' and 1 mL bacterial culture ($OD_{600}$ = 0.5) was seeded onto 92 mm RNAi plates containing 1 mM IPTG. For *sorb-1(RNAi)* 120 L4 larvae, for *vps-4(RNAi)* 240 L4 larvae and for *cogc-2(RNAi)* 200 L4 larvae were transferred onto RNAi plates. Worms were collected in L4 stage after 6 days by washing the plates with MPEG. Worm pellets were subsequently washed using M9 and shock-frozen using liquid nitrogen and kept at -80˚C until extraction.

Lipids were extracted using the BUME method [113]. Briefly, worms were resuspended in 50 µL MeOH and transferred to custom made bead beating tubes. Samples were homogenized at 8000 rpm in a Precellys Bead Beater for 3 times 10 seconds with 20 seconds breaks in between. The additional Cryolys module was used with liquid nitrogen to prevent excessive heating of samples during disruption. 150 µL butanol and 200 µL heptane-ethyl acetate (3:1) was added to each sample sequentially which were then incubated for 1 h at 500 rpm / RT. 200 µL 1% acetic acid was added to each sample followed by centrifugation for 15 min at 13000 rpm / 4˚C. The upper organic phase was transferred to a fresh Eppendorf tube and the lower aqueous phase was re-extracted by the addition of 200 µL heptane-ethyl acetate followed by incubation and centrifugation as described above. The upper organic phase was transferred to the already obtained organic phase. The lower phase was transferred to a new Eppendorf tube and used for metabolomic analyses. Samples were evaporated to dryness and stored at -20˚C. For lipidomics, samples were re-dissolved in 50 µL 65% isopropanol / 35% acetonitrile / 5% $H_2O$, vortexed and 40 µL were transferred to an autosampler vial. The remaining 10 µL were pooled to form a QC sample for the entire study. The precipitated proteins in the aqueous phase were used for determination of protein content using a Bicinchoninic Acid Protein Assay Kit (Sigma-Aldrich, Taufkirchen, Germany).

Lipids were analyzed as previously described [114]. Briefly, lipids were separated on a Waters Acquity UPLC (Waters, Eschborn, Germany) using a Waters Cortecs C18 column (150 mm x 2.1 mm ID, 1.6 µm particle size, Waters, Eschborn Germany) and a linear gradient from 68% eluent A (40% $H_2O$ / 60% acetonitrile, 10 mM ammonium formate and 0.1% formic acid) to 97% eluent B (10% acetonitrile / 90% isopropanol, 10 mM ammonium formate and 0.1% formic acid). Mass spectrometric detection was performed using a Bruker maXis UHR-ToF-MS (Bruker Daltonic, Bremen, Germany) in positive ionization mode using data dependent acquisition to obtain $MS^1$ and $MS^2$ information. Every ten samples, a pooled QC was injected to check performance of the UPLC-UHR-ToF-MS system and used for normalization.

Raw data was processed with Genedata Expressionist for MS 13.0 (Genedata AG, Basel, Switzerland). Preprocessing steps included noise subtraction, m/z recalibration, chromatographic alignment and peak detection and grouping. Data was exported for Genedata Expressionist for MS 13.0 Analyst statistical analysis software and as .xlxs for further investigation. Maximum peak intensities were used for statistical analysis and data was normalized on the protein content of the sample and an intensity drift normalization based on QC samples was used to normalize for the acquisition sequence.

Lipid features that were detected in all pooled QC samples and had a relative standard deviation (RSD) < 30% were further investigated by statistical analysis. 5284 features passed this filter and the different mutants were compared against the wild-type control using Welch test. Lipids with a p-value < 0.05 were considered to be significantly changed.

Lipids were putatively annotated on the $MS^1$ level using an in-house developed database for *C. elegans* lipids and bulk composition from LipidMaps [115], when available. Matching of $MS^2$ spectra against an in-silico database of *C. elegans* lipids and LipidBlast was performed using the masstrixR package [116] (https://github.com/michaelwitting/masstrixR) and only

hits with a forward and reverse matching score > 0.75 were considered. Annotations of interesting biological peaks were manually verified and corrected if necessary.

## High throughput qRT-PCR on single worms using the Biomark system

cDNA from single worms was analyzed on the biomark system using Flex Six IFC. This nano-fluidic chip allows the comparison of 12 target genes across 36 individual worms per genotype. We monitored biological variability in gene expression of targets: endogenous *hsp-6*, *hsp-60* and either *bcSi9* single-copy or *zcIs13* multi-copy transgenes. In addition, we monitored variability in gene expression of three "gold standard" control genes: either non-variable (*hsp-1*), medium variable (*ttr-45*) or highly variable (*nlp-29*). Ct values for all targets were normalized to the average of three housekeeping genes (*cdc-42*, *ire-1* and *pmp-3*).

**Design of qRT-PCR primers.** Primers sets were designed to quantify *C. elegans* post-spliced transcripts. Primer sets were designed to span exon-exon junctions using NCBI Primer Blast software and subsequently blasted against the *C. elegans* genome to test for off-target complementarity. The list of qRT-PCR primers used with their PCR efficiency and coefficient of determination ($R^2$) is shown in S3 Table.

**Quantification of primer efficiency and specificity.** Primers were selected for high PCR efficiency between 90 and 115%. To estimate primer efficiencies, a comprehensive titration of cDNA obtained from 500 ng of Trizol-extracted RNA was prepared within the range of linear amplification using a 1:2 series dilution. Each qRT-PCR reaction contained 1.5 µL of primer mix forward and reverse at 1.6 µM each, 3.5 µL of nuclease free water, 6 µL of 2X Platinum® SYBR® Green qPCR Supermix-UDG (Thermo Fisher Scientific PN 11744–500) and 1 µL of worm DNA lysate diluted or not. The qRT-PCR reactions were run on an iCycler system (Bio-Rad). PCR efficiencies were calculated by plotting the results of the titration of cDNA (Ct values versus log dilution) within the range of linear amplification. The efficiency was defined by the formula 100 x ($10^{(-1/slope)}$/2) with an optimal slope defined as -3.3 ($^{1/3.3}$) = 2.

**Worm synchronization.** Worms were grown at 20˚C and bleach synchronized. 36 worms per genotype were harvested at the L4.8/L4.9 stage based on vulval development [117], at about 48h post L1 plating for WT and about 65h post L1 plating for *fzo-1(tm1133)*.

**Worm lysis for total RNA preparation of single worm RNA.** During harvesting, synchronized worms were individually picked into 10 µL lysis buffer (Power SYBR® Green Cells-to-$C_T$™ kit, Thermo Fisher Scientific) in 8 strip PCR tubes. After harvesting the worms, the 8 strip PCR tubes were freeze-thawed 10 times by transferring tubes from a liquid nitrogen bath into a warm water bath (about 40ºC). Samples were vortexed during 20 minutes on a thermoblock set up at 4ºC. The samples were then quickly spun down and 1 µL of stop solution (Power SYBR Green Cells-to-$C_T$ kit, Thermo Fisher scientific) was added in each tube. The samples were then stored at -80ºC before further processing. Storage time was no more than one week before proceeding to reverse transcription.

**Reverse transcription.** Reverse Transcription PCR (RT-PCR) was performed by adding 5 µL of lysis mix (lysis buffer and stop solution) to 1.25 µL of Reverse Transcription Master Mix (Fluidigm PN 100–6297) into 96 well plates. We included one minus RT control per plate, containing 5 µL of lysis mix and 1.25 µL of RNase free water. Reverse Transcription cycling conditions were 25ºC for 5 min, 42ºC for 30 min and 85ºC for 5 min.

**Pre-amplification.** Pre-amplification was performed according to Fluidigm instruction manual: for every nano-fluidic chip, a pooled primer mix was prepared by adding 1 µL of primer stock (for every target gene to be tested on the chip) to water up to a final volume of 100 µL. Every primer stock contained both reverse and forward primers at a concentration of 50 µM each. A pre-amplification mix was prepared containing for each sample: 1 µL of

PreAmp Master mix (Fluidigm PN 100–5744), 0.5 μL of pooled primer mix and 2.25 μL of nuclease free water. 3.75 μL of pre-amplification mix was then aliquoted in a 96 well-plate. 1.25 μL of cDNA was then added in each well. The samples were mixed by quick vortexing and centrifuged. Pre amplification conditions were the following: 95ºC for 2 min, 10 cycles of denaturation at 95ºC for 15 s followed by annealing/extension at 60ºC for 4 min.

**Exo I treatment and sample dilution.**   To remove unincorporated primers, 2 μL of Exonuclease I mix was added to each pre-amplification reaction. The Exonuclease I mix contained 0.2 μL of Exonuclease I reaction buffer (New England BioLabs), 0.4 μL Exonuclease I at 20 Units/μL (New England BioLabs), and 1.4 μL of nuclease free water. The samples were incubated at 37ºC for 30 min followed by 15 min at 80ºC. The samples were finally diluted 1:5 by adding 18 μL of DNA suspension buffer (10 mM Tris, 0.1 mM EDTA, pH = 8.0, TEKnova PN-T0021).

**Assay Mix preparation.**   For every pair of primers to be tested on the Fluidigm nano-fluidic chip, an assay mix was individually prepared on a 384 well PCR plate (for easier transfer to the Fluidigm nano-fluidic chips), typically the day before the experiment. Each assay mix (for 36 samples) contained 6.25 μL of 2X Assay loading reagent (Fluidigm PN 100–5359), 5 μL of DNA suspension buffer (10 mM Tris, 0.1 mM EDTA, pH = 8.0, TEKnova PN T0021), and 1.25 μL of primer stock (reverse and forward primers at a concentration of 50 μM each). Assay mixes were vortexed during 30 s minimum on a thermoblock set up at 4 ºC and centrifuged for 30 s minimum. 3 μL of each assay mix were loaded onto Flex Six Gene Expression IFC chips (Fluidigm PN 100–6308).

**Sample Mix preparation.**   The samples mixes were prepared at the day of the experiment. 1.8 μL of diluted PreAmp and Exo I treated samples were added to a sample mix containing 2 μL of 2X SsoFast EvaGreen Supermix with Low ROX (Bio-Rad, PN 172–5211) and 0.2 μL of Flex Six Delta Gene Sample Reagent (Fluidigm PN 100–7673). 3 μL of each sample mix was loaded onto Flex Six IFC chips.

**Biomark Run and data clean-up.**   Assay and sample mixes of Flex Six IFCs were loaded using a HX IFC controller (Fluidigm). The nano-fluidic chips were then run on a Biomark HD using the FlexSix Fast PCR+melt protocols. After the run, the data from every well on the plate was checked and cleaned up as following: samples for which all PCRs failed were eliminated. Any well, in which the melting peak temperature of a particular pair of primers was not as expected, was eliminated. It would happen occasionally, presumably when pairs of primers form dimers when target gene concentrations are very low, or from interactions of target primers with other primers in the pooled primer mix. Ct values were then normalized to the average of housekeeping genes and relative mRNA expression levels were calculated using the delta Ct method.

**Determination of "Gold Standard" stable and variable transcripts.**   To validate our single-worm high throughput qRT-PCR method to monitor inter-individual variability in gene expression, we measured the coefficient of variation CV (CV = standard deviation/mean) for fluorescent transcriptional reporters of a stable gene MOC92 *bicIs10(hsp-1p::tagRFP::unc-54 3'UTR)* and of two variable transgenes MOC119 *bicIs12(ttr-45p::tagRFP::ttr45 3'UTR)* (medium variable) and IG274 *frIs7(nlp-29p::GFP; col-12p::DsRed)* (highly variable). We verified that it matches the coefficient of variation calculated from normalized Ct values of endogenous transcripts *hsp-1*, *ttr-45* and *nlp-29* measured in our high-throughput single worm qPCR assay. Synchronized MOC92 and MOC119 transgenic worms were immobilized in M9 containing 3 mM Levamisole and imaged on a Nikon SMZ18 stereo epi-fluorescence microscope, while synchronized IG274 transgenic animals were mounted in 3 mM levamisole on a 2% agarose pad and imaged on a Nikon Ti Eclipse inverted microscope, as the fluorescence levels of the *nlp-29* reporter in IG274 were too low to be imaged on the Nikon SMZ18. The

fluorescence of each individual transgenic worm was quantified using Fiji software, by subtracting the background measurement from fluorescence measurements. The coefficient of variation was determined for synchronized population of day 2 animals (day 2 of adulthood: 74h post L1 plating at 20˚C) for *nlp-29* and *ttr-45* reporters, while it was determined in day 1 synchronized animals (50h post L1 plating at 20˚C) for *hsp-1* reporter. The coefficient of variation is measured as follows:

- *bicIs10(hsp-1p::tagRFP::unc-54 3′UTR)*: 0.09< CV <0.14 (3 biological replicates)

- *bicls12(ttr-45p::tagRFP::ttr45 3'UTR)*: 0.31<CV<0.45 (3 biological replicates)

- *frIs7(nlp-29p::GFP; col-12p::DsRed)*: CV = 1.0 (1 biological replicate)

We observed a good correlation between the coefficient of variation for *hsp-1*, *ttr-45* and *nlp-29* transgenic reporters and the coefficient of variation for endogenous transcripts *hsp-1*, *ttr-45* and *nlp-29* measured by single worm qRT-PCR (S1F Fig).

## Supporting information

**S1 Fig. Comparison of expression levels and inter-individual variability of multi-copy P$_{hsp-6\ mtHSP70}$*gfp (zcIs13)* and single-copy integrated P$_{hsp-6\ mtHSP70}$*gfp (bcSi9)* transgenes. (A)** Brightfield (upper panel) and fluorescence images (lower panel) of L4 larvae expressing P$_{hsp-6}$*gfp (zcIs13)* in wild type (+/+), *spg-7(ad2249)*, *fzo-1(tm1133)* or *drp-1(tm1108)*. Scale bar: 200 μm. **(B)** Brightfield (upper panel) and fluorescence images (lower panel) of L4 larvae expressing P$_{hsp-6}$*gfp (bcSi9)* in wild type (+/+), *spg-7(ad2249)*, *fzo-1(tm1133)* or *drp-1(tm1108)*. Scale bar: 200 μm. **(C)** Quantifications of fluorescence images of panel A (P$_{hsp-6}$*gfp (zcIs13)*) are shown. Each dot represents quantification of 15–20 L4 larvae. Values indicate means ± SD of ≥5 independent measurements. **(D)** Quantifications of fluorescence images of panel B (P$_{hsp-6}$*gfp (bcSi9)*) are shown. Each dot represents quantification of 15–20 L4 larvae. Values indicate means ± SD of ≥4 independent measurements. **(E)** Quantifications of western blot analysis of endogenous HSP-6 levels in wild-type (+/+), *spg-7(ad2249)*, *fzo-1(tm1133)* or *drp-1 (tm1108)* using anti-HSP-6 antibodies. For each genotype, 100 L4 larvae were harvested per experiment for western blot analysis. Values indicate means of relative HSP-6 expression (HSP-6/TUB) ± SD, n = 2. **(F)** Inter-individual variability in gene expression of target genes in *bcSi9* and *zcIs13* in both wild type (+/+) and *fzo-1(tm1133)*. To estimate inter-individual variability in gene expression, the coefficient of variation was calculated from individual mRNA levels obtained from normalized Ct values using the delta Ct method. Inter-individual variability values were normalized such that variability values for *nlp-29* in wild type = 1 *(bcSi9* or *zcIs13)*. Number of individual worms: n = 35 *(bcSi9)*, n = 32 *(bcSi9; fzo-1(tm1133))*, n = 31 *(zcIs13)*, n = 31 *(zcIs13; fzo-1(tm1133))*.
(TIF)

**S2 Fig. RNAi against *vps-4*$^{VPS4}$ and *vps-20*$^{CHMP6}$ suppresses expression of *bcSi9* and induces autophagy in wild type (+/+). (A)** L4 larvae were subjected to *control(RNAi)*, *atfs-1 (RNAi)*, *vps-4(RNAi)* or *vps-20(RNAi)* and the F1 generation was imaged. Each dot represents the quantification of fluorescence intensity of 15–20 L4 larvae. Values indicate means ± SD of 5 independent experiments in duplicates. *$P$<0.05, ***$P$<0.001 using one-way ANOVA with Dunnett's multiple comparison test to *control(RNAi)*. **(B)** P$_{lgg-1}$*gfp::lgg-1* expression of L4 larvae in hypodermal seam cells and intestinal cells upon *control(RNAi)*, *vps-4(RNAi)* or *vps-20 (RNAi)*. Representative images of >30 animals from two independent biological replicates are shown. Scale bar hypodermal seam cells: 5 μm. Scale bar intestinal cells: 20 μm.
(TIF)

**S3 Fig. Knock-down of ESCRT components in body wall muscle cells of wild type and intestinal cells in *fzo-1(tm1133)* does not change mitochondrial morphology. (A)** Fluorescence images of L4 larvae expressing $P_{myo-3}gfp^{mt}$ in wild type (+/+). L4 larvae were subjected to *control(RNAi)*, *atfs-1(RNAi)*, *vps-4(RNAi)*, *vps-20(RNAi)* or *let-363(RNAi)* and the F1 generation was imaged. Scale bar: 10 μm. **(B)** Fluorescence images of L4 larvae expressing $P_{let-858}gfp^{mt}$ in wild type (+/+) or *fzo-1(tm1133)*. L4 larvae were subjected to *control(RNAi)*, *atfs-1 (RNAi)*, *vps-4(RNAi)*, *vps-20(RNAi)* or *let-363(RNAi)* and the F1 generation was imaged. Scale bar: 10 μm.
(TIF)

**S4 Fig. Image segmentation and intensity measurement workflow.** A raw 16-bit image (1) is converted to 8-bit, followed by a background subtraction using the rolling ball algorithm (2). This is followed by the successive application of a minimum (3) and average filter (4). The ImageJ Tubeness plugin generates an image with object curvature scores (5), after which the IsoData autothresholding is applied to generate the binary mask (6). Noise is removed by filtering out particles below a certain size (7) and the final mask is used to define the area in which intensity measurements are obtained (8). Scale bar: 5 μm.
(TIF)

**S5 Fig. Thrashing assay in wild-type and *fzo-1(tm1133)* animals upon induction of autophagy.** Thrashing rate was analyzed by counting body bends of animals swimming for 1 minute in M9 buffer in 3 independent experiments. Each dot represents one L4 larvae. **(A)** Thrashing rates of wild-type (+/+) or *fzo-1(tm1133)* L4 larvae. ****$P<$0.0001 using unpaired two-tailed t-test. n = 30. **(B)** Thrashing rates in wild-type animals upon induction of autophagy. L4 larvae were subjected to *control(RNAi)*, *vps-4(RNAi)*, *vps-20(RNAi)*, *let-363(RNAi)* or *hars-1(RNAi)* and the F1 generation was analyzed. ns: not significant, ****$P<$0.0001 using Kruskal-Wallis test with Dunn's multiple comparison test to *control(RNAi)*. n = 30. **(C)** Thrashing rates in *fzo-1(tm1133)* animals upon induction of autophagy. L4 larvae were subjected to *control(RNAi)*, *vps-4(RNAi)*, *vps-20(RNAi)*, *let-363(RNAi)* or *hars-1(RNAi)* and the F1 generation was analyzed. ns: not significant, ***$P<$0.001 using Kruskal-Wallis test with Dunn's multiple comparison test to *control(RNAi)*. n = 30.
(TIF)

**S6 Fig. RNAi against *vps-4*[VPS4] and *vps-20*[CHMP6] does not suppress *fzo-1(tm1133)*-induced UPR[mt] when diluted with *control(RNAi)* or carried out in one generation from L2 to L4 larvae. (A)** Quantifications of fluorescence images of L4 larvae expressing $P_{hsp-6}gfp$ *(bcSi9)* in *fzo-1(tm1133)*. Each *ESCRT(RNAi)* was diluted 1:1 with *control(RNAi)*. After subtracting the mean fluorescence intensity of wild type (+/+) on *control(RNAi)*, the values were normalized to *fzo-1(tm1133)* on *control(RNAi)*. Each dot represents the quantification of fluorescence intensity of 15–20 L4 larvae. Values indicate means ± SD of 3 independent experiments in duplicates. ns: not significant, using one-way ANOVA with Dunnett's multiple comparison test to *control(RNAi)*. **(B)** Quantifications of fluorescence images of L4 larvae expressing $P_{hsp-6}gfp$ *(bcSi9)* in *fzo-1(tm1133)*. L2 larvae were subjected to *control(RNAi)*, *atfs-1(RNAi)*, *vps-4 (RNAi)* or *vps-20(RNAi)* and the same animals were imaged in L4 stage. After subtracting the mean fluorescence intensity of wild type (+/+) on *control(RNAi)*, the values were normalized to *fzo-1(tm1133)* on *control(RNAi)*. Each dot represents the quantification of fluorescence intensity of 15–20 L4 larvae. Values indicate means ± SD of 4 independent experiments in duplicates. ns: not significant, **$P<$0.01 using Kruskal-Wallis test with Dunn's multiple comparison test to *control(RNAi)*. **(C)** $P_{lgg-1}gfp::lgg-1$ expression of *fzo-1(tm1133)* L4 larvae in

hypodermal seam cells and intestinal cells. L2 larvae were subjected to *control(RNAi)*, *vps-4 (RNAi)* or *vps-20(RNAi)* and the same animals were imaged in L4 stage. Representative images of >60 animals from two independent biological replicates are shown. Scale bar hypodermal seam cells: 5 μm. Scale bar intestinal cells: 20 μm. **(D)** Quantifications of fluorescence images of L4 larvae expressing P*hsp-6*gfp *(bcSi9)* in *fzo-1(tm1133) rrf-3(pk1426)*. L2 larvae were subjected to *control(RNAi)*, *atfs-1(RNAi)*, *vps-4(RNAi)* or *vps-20(RNAi)* and the same animals were imaged in L4 stage. After subtracting the mean fluorescence intensity of wild type (+/+) on *control(RNAi)*, the values were normalized to *fzo-1(tm1133)* on *control(RNAi)*. Each dot represents the quantification of fluorescence intensity of 15–20 L4 larvae. Values indicate means ± SD of 4 independent experiments in duplicates. ns: not significant, ****$P<0.0001$ using one-way ANOVA with Dunnett's multiple comparison test to *control(RNAi)*.
(TIF)

**S7 Fig. Autophagy is induced in *spg-7(ad2249)* animals. (A)** P*lgg-1*gfp::*lgg-1* expression in hypodermal seam cells of wild type (+/+) or *spg-7(ad2249)* L4 larvae. Scale bar: 5 μm. **(B)** Quantification of GFP::LGG-1 foci in hypodermal seam cells from panel A. Each dot represents the average amount of GFP::LGG-1 foci counted from 2–5 seam cells in one animal. n≥18 for each genotype; values indicate means ± SD; **$P<0.01$ using unpaired two-tailed t-test with Welch's correction. **(C)** Nomarski and fluorescent images of the P*sqst-1*sqst-1::*gfp* translational reporter in embryos of wild type (+/+) and *spg-7(ad2249)* animals. As a positive control for a block in autophagy, *unc-51(e369)* was used. Representative images of >60 embryos are shown. Scale bar: 10 μm.
(TIF)

**S8 Fig. Defects in mitochondrial homeostasis lead to major changes in lipid metabolism. (A)** Venn diagrams showing the overlap of lipids up- or downregulated in *fzo-1(tm1133)*, *drp-1(tm1108)* and *spg-7(ad2249)* in comparison to wild type (+/+). **(B)** Total amount of TGs in wild type (+/+), *fzo-1(tm1133)*, *drp-1(tm1108)* and *spg-7(ad2249)* backgrounds. Means ± SD are shown; ns: not significant, *$P<0.05$, ****$P<0.0001$ using Welch test. **(C)** Scatterplot indicating the distribution and changes in the levels of TG species in the different mutants in comparison to wild type (+/+). **(D))** Scatterplot indicating the overlap of the changes in the levels of TG species of *fzo-1(tm1133)* and *drp-1(tm1108)* mutants in comparison to wild type (+/+). **(C)** and **(D)** The x-axis labels the number of carbons (# of C) and the y-axis the number of double bonds (DB) in the acyl sidechains. The size of a dot indicates the number of detected isomers for a specific sum composition. Grey dots represent all detected TGs species and blue and red dots indicate down- (blue) or upregulation (red).
(TIF)

**S9 Fig. Induction of autophagy upon *cogc-2(RNAi)* changes the levels of specific TGs in *fzo-1(tm1133)* mutants. (A)** Scatterplot indicating the distribution and changes in the level of TG species in *fzo-1(tm1133)* mutants in comparison to wild type (+/+). The x-axis labels the number of carbons (# of C) and the y-axis the number of double bonds (DB) in the acyl sidechains. The size of a dot indicates the number of detected isomers for a specific sum composition. Grey dots represent all detected TGs species and blue and red dots indicate down- (blue) or upregulation (red). **(B)** Venn diagram indicating the overlap of TG species downregulated (left panel) or upregulated (right panel) in *fzo-1(tm1133)* and downregulated upon *vps-4 (RNAi)* or *cogc-2(RNAi)*.
(TIF)

**S1 Table. List of genes that suppress *fzo-1(lf)*-induced UPR**mt** and induce autophagy in wild-type animals upon knock-down.** Candidate genes were identified in the primary RNAi-

screen using *fzo-1(tm1133)*, subsequently knocked-down and tested for induction of autophagy and re-screened for UPR<sup>mt</sup> suppression in two different mutant backgrounds: *drp-1 (tm1108)* and *spg-7(ad2249)*.
(XLSX)

**S2 Table. Numerical data of lipidomic experiments. S**ignificantly up- or downregulated lipids in *fzo-1(tm1133)*, *drp-1(tm1108)* or *spg-7(ad2249)* mutants (Sheet 1), significantly up- or downregulated TGs in *fzo-1(tm1133)*, *drp-1(tm1108)* or *spg-7(ad2249)* mutants (Sheet 2) and significantly up- or downregulated TGs in *fzo-1(tm1133)* upon induction of autophagy by *vps-4(RNAi)* or *cogc-2(RNAi)* (Sheet 3). MS[1] annotations, *P*-values and fold change are indicated.
(XLSX)

**S3 Table. List of qRT-PCR primers.** Primers used for qRT-PCR including PCR efficiency and coefficient of determination ($R^2$).
(XLSX)

## Acknowledgments

We thank Eric Lambie, Dejana Mokranjac, the 'Mito Club' and members of the Conradt lab for discussion and comments on the manuscript. We thank M. Bauer, L. Jocham, N. Lebedeva and M. Schwarz for excellent technical support and S. Mitani (National BioResource Project, Tokyo, Japan) for *fzo-1*(*tm1133*), *drp-1(tm1108)* and *atfs-1(tm4525)*. We thank Keith Nehrke and Vincent Galy for *fndc-1(rny-14)*. Some strains were provided by the CGC, which is funded by NIH Office of Research Infrastructure Programs (P40 OD010440).

## Author Contributions

**Conceptualization:** Simon Haeussler, Fabian Köhler, Michael Witting, Stéphane G. Rolland, Laetitia Chauve, Olivia Casanueva, Barbara Conradt.

**Data curation:** Simon Haeussler, Fabian Köhler, Michael Witting.

**Formal analysis:** Simon Haeussler, Fabian Köhler, Michael Witting, Madeleine F. Premm, Christian Fischer, Laetitia Chauve.

**Funding acquisition:** Barbara Conradt.

**Investigation:** Simon Haeussler, Fabian Köhler.

**Methodology:** Simon Haeussler, Fabian Köhler, Michael Witting.

**Project administration:** Barbara Conradt.

**Resources:** Simon Haeussler, Fabian Köhler, Michael Witting.

**Software:** Christian Fischer.

**Validation:** Simon Haeussler, Fabian Köhler, Madeleine F. Premm.

**Visualization:** Simon Haeussler, Fabian Köhler, Michael Witting, Christian Fischer, Laetitia Chauve.

**Writing – original draft:** Simon Haeussler, Fabian Köhler, Michael Witting, Barbara Conradt.

**Writing – review & editing:** Simon Haeussler, Fabian Köhler, Michael Witting, Barbara Conradt.

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
