## [Decision Letter · Decision Letter 0]

22 Aug 2019

Dear Dr Conradt,

Thank you very much for submitting your Research Article entitled 'Autophagy compensates for defects in mitochondrial dynamics' to PLOS Genetics. Your manuscript was fully evaluated at the editorial level and by three independent peer reviewers. The reviewers appreciated the attention to an important problem, but raised some substantial concerns about the current manuscript. Based on the reviews, we will not be able to accept this version of the manuscript, but we would be willing to review again a much-revised version. We cannot, of course, promise publication at that time.

While two of the reviewers found the manuscript to be well written, one thought that the logical flow and conciseness could be improved.  In addition, please be careful to ensure that all experiments presented in the manuscript are properly controlled and that conclusions are justified by appropriate statistical analysis.  Also, at least one reviewer noted that not all of the underlying numerical data is provided (Table S1 for example) – this is a PLOS policy, so please make sure to correct that in the revised version you submit. Your revisions should address the specific points made by each reviewer. We will also require a detailed list of your responses to the review comments and a description of the changes you have made in the manuscript.

If you decide to revise the manuscript for further consideration at PLOS Genetics, please aim to resubmit within the next 60 days, unless it will take extra time to address the concerns of the reviewers, in which case we would appreciate an expected resubmission date by email to plosgenetics@plos.org.

[LINK]

We are sorry that we cannot be more positive about your manuscript at this stage. Please do not hesitate to contact us if you have any concerns or questions.

Yours sincerely,

Gregory P. Copenhaver

Editor-in-Chief

PLOS Genetics

Gregory Barsh

Editor-in-Chief

PLOS Genetics

Reviewer's Responses to Questions

**Comments to the Authors:**

Reviewer #1: This manuscript describes a relatively unexplored relationship between autophagic flux and the regulation of the UPRmt caused by impaired mitochondrial dynamics. A genome-wide RNAi screening approach was used to identify genes that are required for the activation of the UPRmt when mitochondrial fusion is compromised. 299 genetic suppressors of the fzo-1(tm1133) mutant-mediated activation of the UPRmt were identified from this screen of which three were components of the ESCRT. Interestingly, knockdown of these components do not rescue the defects in mitochondrial morphology observed with fzo-1(tm1133) but instead increase mitochondrial membrane potential which is believed to be related to the suppression of the UPRmt observed in this mutant. Using multiple assays, it was shown that RNAi against the ESCRT genes increases autophagy flux in wild-type animals and even more so in the fzo-1(tm1133) mutant. The connection between autophagic flux and the suppression of fzo-1(tm1133) UPRmt activation was validated with knockdown of let-363/mTOR which stimulates autophagy. Remarkably, blocking mitophagy using a pdr-1 mutant did not prevent the suppression of the UPRmt by fzo-1(tm1133). The authors further expanded their findings by testing the remainder of suppressors that had been identified for changes in autophagic flux, of which RNAi of 126 genes increased autophagy (as well as 17 with known autophagy roles totaling 143 suppressors). Finally, while most of the 143 genes also suppressed drp-1(tm1108) induction of the UPRmt, 90 also suppressed the UPRmt activation unrelated to the regulation of mitochondrial dynamics (i.e. the spg-7(ad2249 mutant). Overall the manuscript is well written and explores an interesting correlation between the regulation of the UPRmt with impaired mitochondrial dynamics and autophagic flux, albeit with little mechanism.

1) One of the main interpretations of this manuscript is that increased autophagic flux suppresses the UPRmt resulting from impaired mitochondrial dynamics by restoring mitochondrial homeostasis. However, this conclusion is based primarily on the increase in mitochondrial membrane potential observed with TMRE staining. Are other parameters in mitochondrial function improved as well such as oxygen consumption and ATP production?

2) Related to 1): are there any physiological differences in fzo-1(tm1133) animals when autophagic flux is increased that might support the author’s claim of improved cellular homeostasis? For example thrashing rate, developmental rate, or possibly lifespan?

3) The authors also connect the increase in membrane potential with the regulation of ATFS-1 which is based on mitochondrial import efficiency. Is mitochondrial import improved in fzo-1(tm1133) animals with increased autophagic flux? It would be interesting to examine import efficiency using mitochondrial targeted GFP similar to what was described in Nargund et al. 2012.

4) The authors were not able to show that increased autophagy is necessary for the suppression of the UPRmt in fzo-1(tm1133) animals following ESCRT RNAi because of the embryonic lethality that is observed when the RNAi is performed in the fzo-1(tm1133; unc-51(e369) double mutant background. This is unfortunate since this experiment is rather important to test their central hypothesis. Would a milder form of unc-51 knockdown prevent this associated lethality? Possibly double RNAi against unc-51 and vps-4/20/27 in the fzo-1(tm1133) background? Or introducing a RNAi-hypersensitive mutant (e.g. rrf-3) into the fzo-1(tm1133); unc-51(e69) background which might allow for RNAi to occur in larvae?

5) on p.13, let-363 RNAi suppresses the UPRmt reporters in fzo-1(tm1133) animals to approximately the same level as their control ges-1::GFP. The authors then state however that let-363 RNAi increases TMRE by 19%, supporting their claim that autophagic flux suppresses this UPRmt activation by restoring mitochondrial membrane potential. However, the 19% increase does not appear to be significant based on their quantification in Figure 2F. Is this statement valid then? The same issue also applies to their finding that hars-1 RNAi suppresses ges-1::GFP expression and leads to a 12% increase in TMRE fluorescence which, according to their graph in 2F, is not significant.

Reviewer #2: My review is uploaded as an attachment

Reviewer #3: In the manuscript entitled " Autophagy compensates for defects in mitochondrial dynamics”, Haeussler and colleagues report an analysis of the mitochondrial dynamics using the nematode C. elegans. They first confirm that the depletion of FZO-1 and DRP-1, the respective homologs of mammalian MFN and DRP1, affects the mitochondrial network and show that both mutant induce mitochondrial UPR. The authors performed a genome wide RNAi screen to identify genes which depletion suppresses the UPRmit phenotype of the fzo-1 mutant. Among the candidates, they analyzed further the components of the ESCRT machinery. They show that in fzo-1 mutant there is an induction of autophagy flux and propose that it suppresses the UPRmit. A blockage of autophagy also induces UPRmit. By an RNAi approach, the authors claim that 143 genes are negative regulators of autophagy.

The results presented in the manuscript represent a substantial amount of work, in particular the genome-wide RNAi screen, but several major problems are weakening these results. The main problems concern the absence of clear biological questions driving the experiments, the miss of several controls and the fact that some conclusions are based on single observations and should be confirmed by another independent approach.

1) The result section starts by the description than fzo-1 and drp-1 mutants induce UPRmit.

The two first pages of the results are discussing which reporter of UPRmit is good between a multi copy zcIs13 versus single copy bcSi9, but finally the authors use both strains. This section is not very informative or useful for the rest of the manuscript.

One aspect that is not addressed in the paper is the possibility of tissue-specific effects. It is known that the mitochondrial network is not similar in all C. elegans tissues? Are all the fzo-1 mutant tissues behaving similarly? In figure 1, close up are necessary to see in what tissues hsp-6::GFP and hsp-60::GFP are increased. It is essential to describe what happens in muscle or in epidermis, because these tissues are used to test mitochondrial fragmentation in Fig 2C or TMRE in Fig 2D,E, respectively. Moreover, the quantification of autophagic puncta has been performed within a sub population of epidermal cells (the seam cells).

2) Using an RNAi approach the authors show that the depletion of several ESCRT components suppresses UPRmit in fzo-1 mutant.

In figure 1, some controls are missing. All RNAi should be also shown in the wild-type background and not only in fzo-1(tm1133). What is the efficiency of RNAi ? It has been shown that RNAi against ESCRT are not always recapitulating mutant phenotypes (ref. 36), so mutants could be used to confirm the RNAi phenotype. This is important for ESCRT components which depletion does not present any effect on fzo-1 phenotypes (vps-20 and vps-36).

3) From figure 2, the authors conclude that ESCRT depletion does not improve mitochondrial fragmentation but increases membrane potential. The mitochondrial fragmentation is shown in muscle and the mitochondrial dye TMRE is used in epidermis to analyze the mitochondrial potential membrane. For Fig 2C, pictures of mitoGFP in ESCRT(RNAi) alone should be shown.

TMRE data are not really convincing because alone it does not allow deciphering between a change in the membrane potential and a change in the mitochondrial mass. One possibility is to combine TMRE with another mitochondrial dye independent of MMP, or a GFP marker of the mitochondria. It is surprising that fzo-1 mutant have almost no staining with TMRE in Fi g2D. The measure of the oxygen consumption rate should be also perform to confirm these data.

4) The authors convincingly show that there is an increase of autophagic flux in fzo-1 animals. Fig 3C is too small and data are not quantified.

ESCRT(RNAi); fzo-1 animals present a further increase in autophagy. It supports previously published data showing that ESCRT depletion induces autophagic flux but does not prove that there is a causal link between FZO-1 and ESCRT. In Fig 3E, controls for ESCRT(RNAi) alone are not shown.

5) Using let-363(RNAi) (TOR homolog), the authors conclude that an autophagy induction suppresses the mitochondrial UPR in fzo-1 mutant. mTOR is involved in regulating autophagy but also in many others processes that can affect homeostasis and let-363 animals have many phenotypes. It could be useful to show that a blockage of autophagy flux (atg-5, atg-7 …) in let-363 mutant blocks the effect on UPRmit.

6) The paragraph “Depletion of ESCRT components in fzo-1(tm1133) animals with a block in autophagy results in embryonic lethality” is not informative because the authors could not test whether the increase of autophagic flux in ESCRT depleted fzo-1 animals is responsible of the improvement of mitochondrial UPR. Then, they tested RNAi against one pathway involved in mitophagy. However, the authors have not demonstrated that there is mitophagy occurring in fzo-1 mutants or ESCRT(RNAi); fzo-1. Moreover, there are others mitophagic pathways parallel to PDR-1 which have not been tested (Fundc1). The conclusion of this section is rather weak.

7) The authors show data indicating that blocking autophagy induces UPRmit but does not further increase UPRmit in fzo-1. It would be interesting to test whether it is due to an accumulation of damages that are not removed or an active induction.

Conversely depleting ATFS-1 does not modify autophagic response of fzo-1 animals. Unfortunately, the effect of overexpression could not been studied.

8) The last part of the manuscript presents a combination of data mining and RNAi to identify 143 candidates that the authors qualify of “negative regulators of autophagy”. This is affirmed on the observation that RNAi of these genes results in increase of GFP::LGG-1 dots in the seam cells and no accumulation of SQST-1. Six genes among the 143 candidates have been further tested for autophagy induction after RNAi depletion.

This approach shows that autophagy is potentially increased upon RNAi treatments, but it could be a very indirect consequence. Demonstrating that the overexpression of these proteins decrease the autophagic flux is necessary to conclude that these candidates are bona fide negative regulators.

9) In the discussion, an interesting hypothesis proposes that the modifications of mitochondrial networks (fzo-1 and drp-1) induce the UPRmit and then a metabolic change. Do the authors have any data about a potential metabolic shift?

Authors should discuss the potential mechanism(s) of action of ESCRTs on autophagy and mitochondrial dynamics. ESCRT defective mutant triggers an adaptative autophagic flux, but recent data have shown that ESCRT are involved in autophagosome closure (Takahashi Nature com 2018; Zhou JCB 2019).These references should be added and discussed.

Minor points

- For laudable pedagogic reason the authors propose a combination of C. elegans and human names for genes and proteins, but it does not respect the C. elegans nomenclature and in particular cases complicate the comprehension (Phsp-6 mtHSP70gfp).

- C. elegans Vps-27 official name is Hgrs.

- LGG-1 is the homolog of the GABARAP family rather than the LC3

**Have all data underlying the figures and results presented in the manuscript been provided?**

Reviewer #1: Yes

Reviewer #2: No: Numerical data underlying Table S1 is not provided.

Reviewer #3: None

PLOS authors have the option to publish the peer review history of their article (what does this mean?). If published, this will include your full peer review and any attached files.

Reviewer #1: No

Reviewer #2: Yes: Keith Nehrke

Reviewer #3: No

---

## [Decision Letter · Decision Letter 1]

28 Jan 2020

Dear Dr Conradt,

We are pleased to inform you that your manuscript entitled "Autophagy compensates for defects in mitochondrial dynamics" has been editorially accepted for publication in PLOS Genetics. Congratulations!

Reviewers #1 and #2 are now satisfied with the manuscript and were quite complementary of the effort to address their prior comments.  Reviewer #3 has some remaining concerns.  I believe these can be addressed, if necessary (see below), textually while preparing the final draft of your manuscript for the production team (the editorial team will not need to re-evaluate the manuscript).  Regarding the specific issues raised by Reviewer #3:

Close-up of UPRmt reporters in the intestine – the other reviewers did not see this as necessary, so I won’t make acceptance contingent on it.  If you now have additional images of this type that you’d like to add, feel free to include them.Description of RNAi depletion – please address this in the text.Title – please consider whether the title needs to be modified.  Ultimately, I’ll leave this up to you, but in my experience, over-reaching is often more deleterious than circumspection. Figure 7 & S8 – please add quantitative descriptions as appropriate. 

Yours sincerely,

Gregory P. Copenhaver

Editor-in-Chief

PLOS Genetics

Gregory Barsh

Editor-in-Chief

PLOS Genetics

Comments from the reviewers (if applicable):

Reviewer's Responses to Questions

**Comments to the Authors:**

Reviewer #1: I am satisfied with the revision experiments that were performed. There was a clear effort made for those experiments that were unsuccessful, which were not possible due to technical considerations.

Reviewer #2: In general, I feel that the rebuttal is responsive and addresses many of the reviewers' concerns. It is unfortunate that several attempts to provide additional data met with technical challenges, but the authors made a good faith effort, and the overall conclusions have been strengthened in the revised version. I also appreciated their ending with the lipid story, as it serves as an appropriate segue into future studies. This is a complicated story, but one well-worth spending time dissecting.

Reviewer #3: The revision of the manuscript by Haeussler and colleagues has improved several aspects and questions that I mentioned on the original version. A number of controls have been added and the authors have better explained some technical aspects that were unclear, e.g. the quantification of mitochondrial staining.

There are still some concerns about the replies to specific points and a comment about the new results:

- Point1. The justification for not showing close-up of UPRmt reporters in the intestine and the epidermis is not really convincing.

- Point 2 about RNAi efficiency has not be really addressed. It is important for genes whose depletion by RNAi does not result in the suppression of UPRmt. The negative result for ESCRT-II vps-22; vps-36 could be due to RNAi inefficiency and conclusion in line 220 is inappropriate.

- Point 5. There is a correlation between induction of autophagy and suppression of UPRmit, but unfortunately, the authors did not succeed to show that blocking autophagy inhibits the suppression of UPRmt observed in LET-363 or ESCRT mutants. In absence of an evidence, the tittle of the manuscript and the interpretation of the results should be more careful.

- The new data described in the lipidomic approach are interesting but what are the quantitative effects (Panel B in figure 7 and panels C and D in figure S8) ? An indication about fold change threshold for up and down regulated lipids is necessary.

**Have all data underlying the figures and results presented in the manuscript been provided?**

Reviewer #1: None

Reviewer #2: Yes

Reviewer #3: Yes

PLOS authors have the option to publish the peer review history of their article (what does this mean?). If published, this will include your full peer review and any attached files.

Reviewer #1: No

Reviewer #2: Yes: Keith Nehrke

Reviewer #3: No

**Data Deposition**

http://datadryad.org/submit?journalID=pgenetics&manu=PGENETICS-D-19-01004R1

**Press Queries**

---

## [Editor Report · Acceptance letter]

11 Mar 2020

PGENETICS-D-19-01004R1 

Autophagy compensates for defects in mitochondrial dynamics 

Dear Dr Conradt, 

We are pleased to inform you that your manuscript entitled "Autophagy compensates for defects in mitochondrial dynamics" has been formally accepted for publication in PLOS Genetics! Your manuscript is now with our production department and you will be notified of the publication date in due course.

With kind regards,

Kaitlin Butler

PLOS Genetics

On behalf of:
